# β2-subunit alternative splicing stabilizes Cav2.3 Ca²⁺ channel activity during continuous midbrain dopamine neuron-like activity

Anita Siller[1], Nadja T Hofer[1], Giulia Tomagra[2], Nicole Burkert[3], Simon Hess[4], Julia Benkert[3], Aisylu Gaifullina[3], Desiree Spaich[3], Johanna Duda[3], Christina Poetschke[3], Kristina Vilusic[1], Eva Maria Fritz[1], Toni Schneider[5], Peter Kloppenburg[4], Birgit Liss[3,6], Valentina Carabelli[2], Emilio Carbone[2], Nadine Jasmin Ortner[1]*, Jörg Striessnig[1]*

[1]Department of Pharmacology and Toxicology, Institute of Pharmacy, Center for Molecular Biosciences Innsbruck, University of Innsbruck, Innsbruck, Austria; [2]Department of Drug Science, NIS Centre, University of Torino, Torino, Italy; [3]Institute of Applied Physiology, University of Ulm, Ulm, Germany, Ulm, Germany; [4]Institute for Zoology, Biocenter, University of Cologne, Cologne, Germany; [5]Institute of Neurophysiology, University of Cologne, Cologne, Germany; [6]Linacre College & New College, University of Oxford, Oxford, United Kingdom

**\*For correspondence:**
nadine.ortner@uibk.ac.at (NJO);
joerg.striessnig@uibk.ac.at (JS)

**Abstract** In dopaminergic (DA) *Substantia nigra* (SN) neurons Cav2.3 R-type Ca²⁺-currents contribute to somatodendritic Ca²⁺-oscillations. This activity may contribute to the selective degeneration of these neurons in Parkinson's disease (PD) since Cav2.3-knockout is neuroprotective in a PD mouse model. Here, we show that in tsA-201-cells the membrane-anchored β2-splice variants β2a and β2e are required to stabilize Cav2.3 gating properties allowing sustained Cav2.3 availability during simulated pacemaking and enhanced Ca²⁺-currents during bursts. We confirmed the expression of β2a- and β2e-subunit transcripts in the mouse SN and in identified SN DA neurons. Patch-clamp recordings of mouse DA midbrain neurons in culture and SN DA neurons in brain slices revealed SNX-482-sensitive R-type Ca²⁺-currents with voltage-dependent gating properties that suggest modulation by β2a- and/or β2e-subunits. Thus, β-subunit alternative splicing may prevent a fraction of Cav2.3 channels from inactivation in continuously active, highly vulnerable SN DA neurons, thereby also supporting Ca²⁺ signals contributing to the (patho)physiological role of Cav2.3 channels in PD.

## Editor's evaluation

This study finds that voltage-gated calcium channel auxiliary β2a and β2e splice variants confer gating properties on Cav2.3 channels that enable them to contribute sustained Ca²⁺ influx during pacemaking in substantia nigra dopaminergic neurons. This sustained Ca²⁺ influx may contribute to the selective vulnerability of substantia nigra dopaminergic neurons to neurodegeneration in Parkinson's disease. The work will be of great interest to ion channel biophysicists and neuroscientists interested in mechanisms of neurodegeneration.

## Introduction

Parkinson's disease (PD) is one of the most common neurodegenerative disorders. Its motor symptoms are characterized by progressive degeneration of dopamine (DA)-releasing neurons in the *Substantia nigra* (SN), while neighboring DA neurons in the ventral tegmental area (VTA) remain largely unaffected (*Damier et al., 1999*; *Giguère et al., 2018*; *Surmeier et al., 2017*). Current PD therapy is only symptomatic and primarily based on the substitution of striatal DA by administration of L-DOPA or dopamine D2 receptor agonists. Unfortunately, none of the existing therapeutic approaches for PD patients is disease-modifying and can prevent disease progression (for review see *Liss and Striessnig, 2019*; *Surmeier et al., 2011*; *Surmeier et al., 2017*).

The development of novel neuroprotective strategies for the treatment of early PD requires the understanding of the cellular mechanisms responsible for the high vulnerability of SN DA neurons. Among these mechanisms, elevated metabolic stress appears to play a central role (for review see *Liss and Striessnig, 2019*), eventually triggering lysosomal, proteasomal and mitochondrial dysfunction (*Burbulla et al., 2017*; *Surmeier et al., 2017*). Intrinsic physiological properties of SN DA neurons, in particular increased cytosolic DA levels and high energy demand due to large axonal arborization favor metabolic stress (*Bolam and Pissadaki, 2012*; *Liss and Striessnig, 2019*). In addition, these neurons must handle a constant intracellular $Ca^{2+}$-load resulting from dendritic and somatic $Ca^{2+}$-oscillations triggered during their continuous electrical activity (*Ortner et al., 2017*; *Surmeier et al., 2011*). Dendritic $Ca^{2+}$-transients largely depend on the activity of voltage-gated $Ca^{2+}$ channels, in particular Cav1.3 L-type (LTCCs) and T-type channels (*Guzman et al., 2018*). Cav1.3 channels can activate at subthreshold membrane potentials (*Koschak et al., 2001*; *Lieb et al., 2014*; *Xu and Lipscombe, 2001*) and do not completely inactivate during continuous pacemaking activity (*Guzman et al., 2018*; *Guzman et al., 2009*; *Ortner et al., 2017*). Some, but not all, in vivo studies (*Liss and Striessnig, 2019*) showed neuroprotection by the systemic administration of dihydropyridine (DHP) LTCC blockers in 6-OHDA and MPTP animal models of PD thus further supporting a role of LTCCs as potential neuroprotective drug target. Based on these preclinical data and supporting observational clinical evidence (*Liss and Striessnig, 2019*), the neuroprotective potential of the DHP isradipine (ISR) was tested in a double-blind, placebo-controlled, parallel-group phase 3 clinical trial ("STEADY-PD III", NCT02168842; *Biglan et al., 2017*). This trial reported no evidence for neuroprotection by ISR. Several explanations have been offered for this negative outcome (*Investigators, 2020*). One likely explanation is that voltage-gated $Ca^{2+}$-channels (Cavs) other than LTCCs also contribute to $Ca^{2+}$-transients in SN DA neurons. This is supported by the observation that only about 50% of the $Ca^{2+}$-transients are blocked by ISR in the dendrites of SN DA neurons (*Guzman et al., 2018*) and that action potential-associated $Ca^{2+}$-transients in the soma appear to be even resistant to ISR (*Ortner et al., 2017*). Therefore, in addition to L-type, other types of Cavs expressed in SN DA neurons (*Branch et al., 2014*; *Evans et al., 2017*; *Philippart et al., 2016*) may also contribute to $Ca^{2+}$-induced metabolic stress in SN DA neurons. Cav2.3 (R-type) $Ca^{2+}$-channels are very promising candidates. We have recently shown that SN DA neurons in mice lacking Cav2.3 channels were fully protected from neurodegeneration in the chronic MPTP-model of PD. Moreover, we found that Cav2.3 is the most abundant Cav expressed in SN DA neurons, and substantially contributes to activity-related somatic $Ca^{2+}$-oscillations (*Benkert et al., 2019*). These findings make Cav2.3 R-type $Ca^{2+}$ channels a promising target for neuroprotection in PD.

SN DA neurons are spontaneously active, pacemaking neurons, either firing in a low-frequency single-spike mode or transiently in a high-frequency burst mode (*Grace and Bunney, 1984*; *Paladini and Roeper, 2014*). During regular pacemaking their membrane potential is, on average, rather depolarized ranging from about –70 mV after an action potential (AP) to about –40 mV at firing threshold (*Gantz et al., 2018*; *Guzman et al., 2018*; *Ortner et al., 2017*). The contribution of a particular Cav channel to $Ca^{2+}$-entry is largely determined by its steady-state inactivation properties, which determines its availability at these depolarized voltages. Therefore, in SN DA neurons steady-state inactivation of Cav2.3 channels must occur within a voltage-range preventing inactivation of a substantial fraction of channels during the positive operating range of these continuously firing neurons. However, Cav2.3 α1-subunits have originally been cloned as a low-voltage-gated channel with a negative steady-state inactivation voltage-range ($V_{0.5,inact}$ –78 mV; *Soong et al., 1993*) with almost complete inactivation at voltages positive to –50 mV. Interestingly, SNX-482-sensitive R-type currents mediated by Cav2.3 α1 subunits (*Newcomb et al., 1998*) recorded from different neuronal

cell types reveal a wide voltage-range of steady-state inactivation despite similar recording conditions (5–10 mM $Ba^{2+}$ as charge carrier): half-maximal voltages for steady-state inactivation ($V_{0.5,inact}$) range from –90 to –70 mV in cerebellar, cortical and myenteric neurons (**Bian et al., 2004**; **Sochivko et al., 2002**; **Tottene et al., 2000**) to –58 mV in neurohypophyseal terminals (**Wang et al., 1999**) and to even as positive as –40 mV in GnRH-releasing neurons or GT1-7 cells (**Kato et al., 2003**; **Watanabe et al., 2004**). This raises the important question about potential molecular mechanisms accounting for this heterogeneity and its role for stabilization of R-type currents in SN DA neurons.

While alternative splicing of Cav2.3 α1 subunits is unlikely to account for such large shifts in steady-state inactivation (**Pereverzev et al., 2002**), accessory β-subunits appear as obvious candidates for several reasons. First, like for other high-voltage-activated $Ca^{2+}$ channels (including Cav2.1 and Cav2.2, **Liu et al., 1996**; **Scott et al., 1996**) they form part of the Cav2.3 channel complex and thus affect its membrane targeting and gating properties (**Zamponi et al., 2015**). Second, previous work has shown that β-subunits tightly control $V_{0.5,inact}$ of Cav2 channels, including Cav2.3. While all cytosolic β-subunit isoforms (β1 – β4 and cytosolic splice variants of β2) induce steady-state inactivation at negative potentials, being almost complete at voltages positive to –50 mV, the membrane-anchored β2 subunit splice variants β2a and β2e induce strong positive shifts of $V_{0.5,inact}$ of Cav2.3 channels and slow the time course of voltage-dependent inactivation (**Jones et al., 1998**; **Pereverzev et al., 2002**; **Soong et al., 1993**; **Williams et al., 1994**; **Yasuda et al., 2004**). However, a specific physiological role for this modulatory effect has so far not been reported and the relative abundance of β2a and β2e subunits in neurons, including SN DA neurons, is unknown.

Here, we directly addressed the question if association of Cav2.3 channels with β2a or β2e but not with cytosolic β-subunits can prevent channel inactivation during SN DA activity patterns and could therefore serve as a mechanism allowing these channels to contribute to SN DA function. When co-expressed together with auxiliary α2δ1-subunits in tsA-201 cells we indeed found that during simulated SN DA neuron-like continuous tonic pacemaking or brief burst activity (**Ortner et al., 2017**) Cav2.3 channels remained active only when associated with β2a or β2e, but not the cytosolic β3 and β4 subunits. In contrast, steady-state inactivation of Cav1.3 channels was largely unaffected by β2a, suggesting that Cav1.3 availability is much less dependent on the presence of β2a subunits. Using RNAscope we confirmed the presence of both β2a and β2e transcripts in identified mouse SN and VTA DA neurons and quantitative PCR analysis showed that β2-subunits represent about 25% of all β-subunit transcripts in the mouse SN and VTA with about 50% comprising β2a and β2e variants. In patch-clamp recordings of SN DA neurons in mouse brain slices, we detected SNX-482-sensitive R-type $Ca^{2+}$-currents with voltage-dependent inactivation properties suggesting a contribution of β2a- or β2e-stabilized Cav2.3-currents. Recordings in cultured DA midbrain neurons confirmed R-type current activity during the pacemaking cycle. Taken together, our data further support a role of Cav2.3 in SN DA neuron $Ca^{2+}$-signaling and imply a potential (patho)physiological role of β-subunit alternative splicing.

## Results

To explore how Cav2.3 channels can contribute to DA neuron $Ca^{2+}$-entry during sustained neuronal activity we expressed Cav2.3 α1-subunits together with its accessory α2δ1 and different β-subunits in tsA-201 cells under near-physiological conditions (**Figure 1**). For this purpose, we employed physiological extracellular $Ca^{2+}$ (2 mM), weak intracellular $Ca^{2+}$-buffering (0.5 mM EGTA, see methods), and, in addition to square pulse protocols, used typical AP waveforms previously recorded from SN DA neurons in mouse midbrain slices (2.5 Hz) or simulated bursts as command voltages as described (**Ortner et al., 2017**; see methods). Moreover, we specifically employed the Cav2.3e α1-subunit splice variant for our recordings, which among the six major Cav2.3 α1-subunit splice variants, was the only one detected in UV laser-microdissected mouse SN DA neurons in experiments using a qualitative single-cell RT-qPCR approach (**Figure 1—figure supplement 1A, B**).

### β-subunit isoform-dependent regulation of Cav2.3 channel gating

We first used standard square pulse protocols to quantify the effect of different β-subunits on the voltage-dependence of channel gating under our experimental conditions. In addition to β3, β4, and β2a analyzed in previous studies (**Jones et al., 1998**; **Yasuda et al., 2004**), we also included

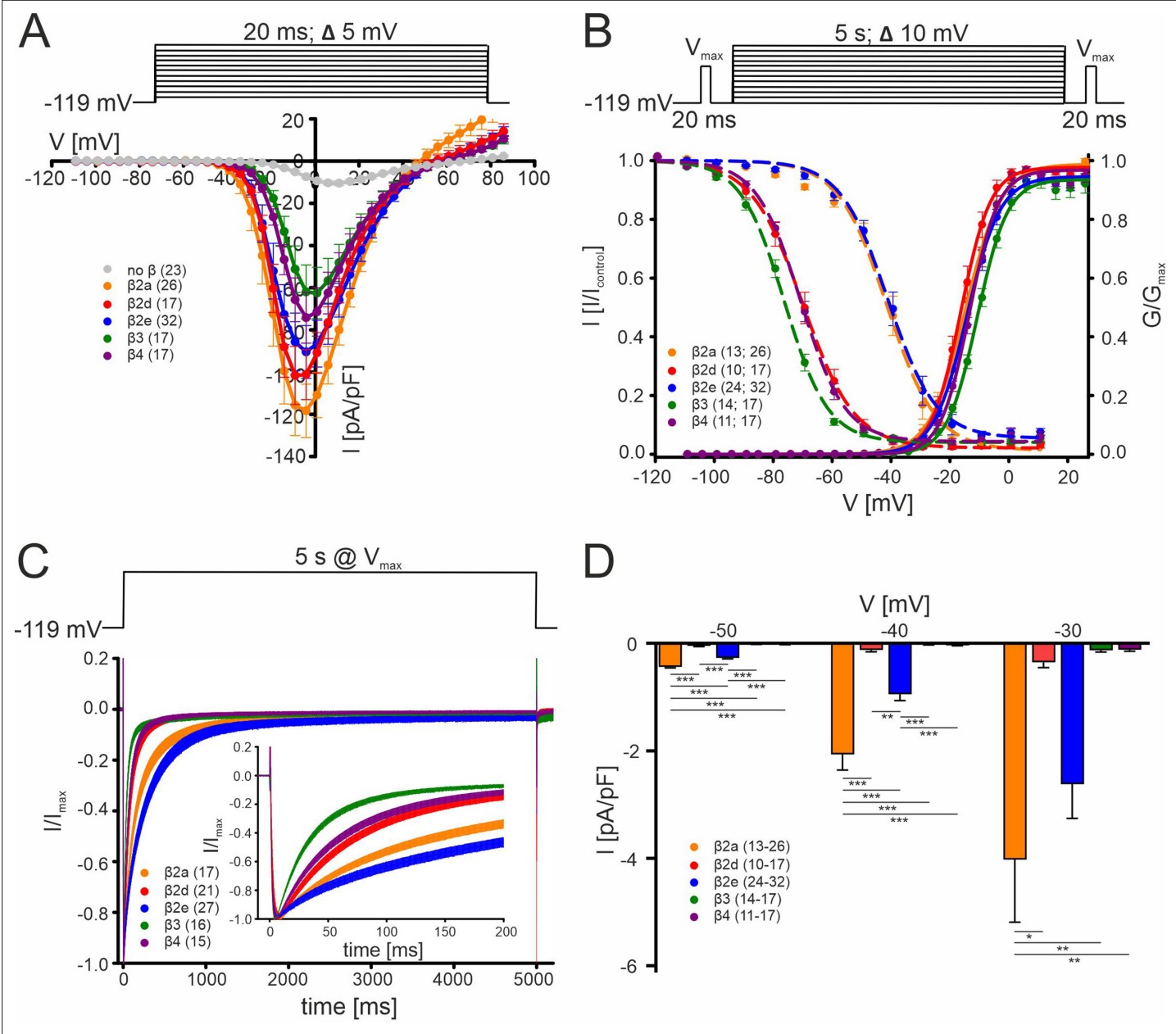

**Figure 1.** Biophysical properties of Cav2.3 channels co-transfected with different β-subunits (and α2δ1) in tsA-201 cells. (**A**) Current densities (pA/pF) with or without (gray) co-transfection of indicated β-subunits. Color code and n-numbers are given in the graphs. (**B**) Voltage-dependence of steady-state activation (normalized conductance G, right axis, solid lines) and inactivation (normalized $I_{Ca}$ of test pulses, left axis, dashed lines, left n-numbers in parentheses). (**C**) Inactivation time course during 5 s depolarizing pulses to $V_{max}$ starting from a holding potential of –119 mV. Inset shows the first 200 ms of the 5 s pulse. Respective stimulation protocols are shown above each graph. The curves represent the means ± SEM for the indicated number of experiments (N = β2a: 5; β2d, β2e: 2; β3, β4, no β: 3). For statistics see **Table 1**. $V_{max}$, voltage of maximal inward current. (**D**) Window currents measured in the presence of the indicated β-subunits were calculated by multiplying mean current densities (pA/pF) of I-V-relationships by the fractional current inactivation from steady-state inactivation curves at the indicated voltages. Data represent the means ± SEM for the indicated number of experiments (N = β2a: 5; β2d, β2e: 2; β3, β4: 3). Statistical significance was determined using one-way ANOVA with Bonferroni post-hoc test and is indicated: *** p<0.001; ** p<0.01; * p<0.05. Source data provided in **Figure 1—source data 1**.

The online version of this article includes the following source data and figure supplement(s) for figure 1:

**Source data 1.** Source data for data shown in **Figure 1**.

**Figure supplement 1.** Determination of the expressed Cav2.3 splice variants in SN DA neurons.

**Figure supplement 2.** Effect of β2a palmitoylation on the biophysical properties of Cav2.3 and Cav1.3 Ca²⁺ channels in tsA-201 cells.

**Figure supplement 2—source data 1.** Source data for data shown in **Figure 1—figure supplement 2**.

**Table 1.** Voltage-dependence of activation and inactivation, and time course of inactivation of Cav2.3 co-transfected with α2 δ 1 and different β subunits in tsA-201 cells.

All values are given as means ± SEM for the indicated number of experiments (N = β2a: 5; $_{C3S/C4S}$β2a, β2d, β2e: 2; β3, β4, no β: 3). Voltage-dependence of gating: $V_{0.5}$, Half-maximal activation voltage; k, slope factor; $V_{rev}$, estimated reversal potential; act thresh, activation threshold; $V_{0.5,inact}$, half-maximal inactivation voltage; $k_{inact}$, inactivation slope factor; plateau, remaining non-inactivating current. Near physiological recording conditions (2 mM $Ca^{2+}$, low 0.5 mM EGTA $Ca^{2+}$ buffering) and calculation of the parameters of voltage-dependence of activation and inactivation are described in Materials and Methods. Statistical significance was determined using one-way ANOVA with Bonferroni post-hoc test ($V_{0.5}$, $V_{rev}$, $V_{0.5,inact}$, $k_{inact}$, plateau) or Kruskal-Wallis followed by Dunn's multiple comparison test (k, act thresh, current density). Statistical significances of post-hoc tests are indicated for comparison vs. β2a (*, **, ***), vs. β2e (#, ##, ###), vs. β3 (§, §§, §§§) and vs. no β (+, ++, +++): p<0.05, p<0.01, p<0.001. Inactivation time course: The r values represent the fraction of $I_{Ca}$ remaining after 50, 100, 250, 500, 1000. or 5000 ms during a 5 s pulse to $V_{max}$ (voltage of maximal inward current). Statistical significance was determined using one-way ANOVA with Bonferroni post-hoc test (r100) or Kruskal-Wallis followed by Dunn's multiple comparison test (r50, r250, r500, r1000, r5000). Statistical significances of post hoc tests are indicated for comparison vs. β2a (*, **, ***), vs. β2e (#, ##, ###), vs. β3 (§, §§, §§§) or vs β4 (%, %%, %%%): p<0.05, p<0.01, p<0.001.

Source data provided in *Table 1—source data 1*.

| β-subunit | Cav2.3 - Activation 2 mM $Ca^{2+}$ | | | | | | Cav2.3 - Inactivation 2 mM $Ca^{2+}$ | | | |
|---|---|---|---|---|---|---|---|---|---|---|
| | $V_{0.5}$ [mV] | k [mV] | $V_{rev}$ [mV] | act thresh [mV] | current density [pA/pF] | n | $V_{0.5, inact}$ [mV] | $k_{inact}$ [mV] | plateau [%] | n |
| no β | | | | | −10.7±1.1 | 23 | | | | |
| β2a | −14.8±1.2 | 4.8±0.2 | 37.1±0.9 | −32.0±0.9 | −130.0+++±15.2 | 26 | −40.6±1.6 | 7.3±0.5 | 0.4±1.1 | 13 |
| $_{C3S/C4S}$β2a | −13.9±1.7 | 4.7±0.3 | 38.2±1.1 | −30.9±0.7 | −105.7+++±21.4 | 12 | −62.6***/###/§§§±1.6 | 8.3##±0.4 | 3.8±1.6 | 9 |
| β2d | −15.9§±1.1 | 4.6±0.2 | 37.2±1.1 | −32.1§±0.7 | −107.7+++±13.4 | 17 | −69.9***/###±2.0 | 8.2##±0.2 | 2.0±0.7 | 10 |
| β2e | −14.1±0.9 | 4.8±0.2 | 40.3*±0.6 | −31.5±0.7 | −96.8+++±14.3 | 32 | −39.9±1.9 | 6.5±0.3 | 3.8±1.2 | 24 |
| β3 | −10.5±0.7 | 5.2±0.2 | 39.8±0.9 | −29.2±0.7 | −64.6++±12.8 | 17 | −76.2***/###±1.0 | 7.0±0.1 | 4.1±0.7 | 14 |
| β4 | −13.1±0.7 | 5.0±0.2 | 38.5±0.5 | −31.0±0.5 | −74.8+++±11.9 | 17 | −70.3***/###±1.2 | 7.5±0.1 | 2.4±0.4 | 11 |

**Cav2.3–5 s Inactivation time course 2 mM $Ca^{2+}$**

| β-subunit | r50 [%] | r100 [%] | r250 [%] | r500 [%] | r1000 [%] | r5000 [%] | n |
|---|---|---|---|---|---|---|---|
| β2a | 71.3±2.1 | 52.4±2.8 | 27.8±2.9 | 14.3±2.3 | 6.5±1.2 | 1.3±0.3 | 17 |
| $_{C3S/C4S}$β2a | 67.4§§§/%±5.2 | 47.3#/§§§/%%±6.4 | 23.4§±5.3 | 11.6±3.3 | 6.2±1.7 | 3.4±1.1 | 11 |
| β2d | 57.4*/###/§§§±2.8 | 32.6***/###/§§±2.4 | 11.3*/###±1.7 | 4.4**/###±0.9 | 2.4**/###±0.6 | 1.7±0.5 | 21 |
| β2e | 77.5±2.9 | 64.2±3.3 | 41.1±3.2 | 22.3±2.8 | 10.1±2.0 | 3.3±0.8 | 27 |
| β3 | 32.9***/###±2.9 | 14.2***/###±1.5 | 6.1***/###±1.0 | 4.4**/###±0.7 | 2.9#±0.5 | 1.6±0.3 | 16 |
| β4 | 50.3***/###/§§±2.7 | 27.1***/###±2.5 | 8.6**/###±1.4 | 3.7**/###±0.8 | 2.0**/###±0.5 | 1.0±0.2 | 15 |

The online version of this article includes the following source data for table 1:

**Source data 1.** Source data for data shown in *Table 1*.

membrane-anchored β2e and β2d, a cytosolic β2 splice variant, in our head-to-head comparison (*Figure 1*). In agreement with earlier studies (*Jones et al., 1998*; *Yasuda et al., 2004*), we found that recombinant Cav2.3 channels when associated with cytosolic β3 or β4 subunits inactivate at negative voltages ($V_{0.5,inact}$ < –70 mV) and with a rapid inactivation time course (≥50% within 50 ms, *Figure 1B–C*, *Table 1*). The cytosolic β2d splice variant also showed a gating behavior very similar to β4. In contrast, β2a, which is anchored to the plasma membrane through N-terminal palmitoylation, slowed the inactivation time course and shifted Cav2.3 voltage-dependent inactivation by ~30 mV to more positive potentials compared to β4 (and β3). Likewise, with co-expressed β2e, which is membrane anchored through N-terminal phospholipid interaction (*Buraei and Yang, 2010*; *Miranda-Laferte et al., 2014*), gating properties were similar to β2a-containing Cav2.3 channel complexes (*Figure 1A–C*, *Table 1*). In the case of β2a, the shift in the voltage-dependence of inactivation, but not the slowing of the inactivation time course, was largely dependent on N-terminal palmitoylation as shown with the palmitoylation-deficient mutant $_{C3S/C4S}$β2a (*Figure 1—figure supplement 2*). The voltage-dependence of activation was similar for all tested β-subunits (*Figure 1A*, *Table 1*). Due to the voltage-dependent inactivation at more depolarized potentials and the resulting overlap of the steady-state activation

and inactivation curves (*Figure 1B*), β2a and β2e subunits induce window currents, i.e. steady-state $Ca^{2+}$ influx, at negative potentials (*Figure 1D*). Our findings confirm and extend previous observations that membrane-anchored β2 subunits give rise to slowly inactivating Cav2.3 currents available at more positive voltages that could participate in the formation of such SNX-482-sensitive current components in neurons.

Since Cav2.3 α1-subunits have also been reported to mediate $I_{Ca}$ even when expressed in the absence of β-subunits (*Jones et al., 1998*; *Yasuda et al., 2004*), we also tested to which extent β-subunit modulated channels are expected to contribute to Cav2.3 currents. In our experiments, all five tested β-subunits (β2a, β2d, β2e, β3, β4) caused a robust and highly significant (6–12-fold; *Figure 1A*, *Table 1*) increase in current densities. This implies that β-associated Cav2.3 channels contribute more to overall Cav2.3-mediated currents than channel complexes devoid of β-subunits and thus can be subject to differential modulation by β-subunits.

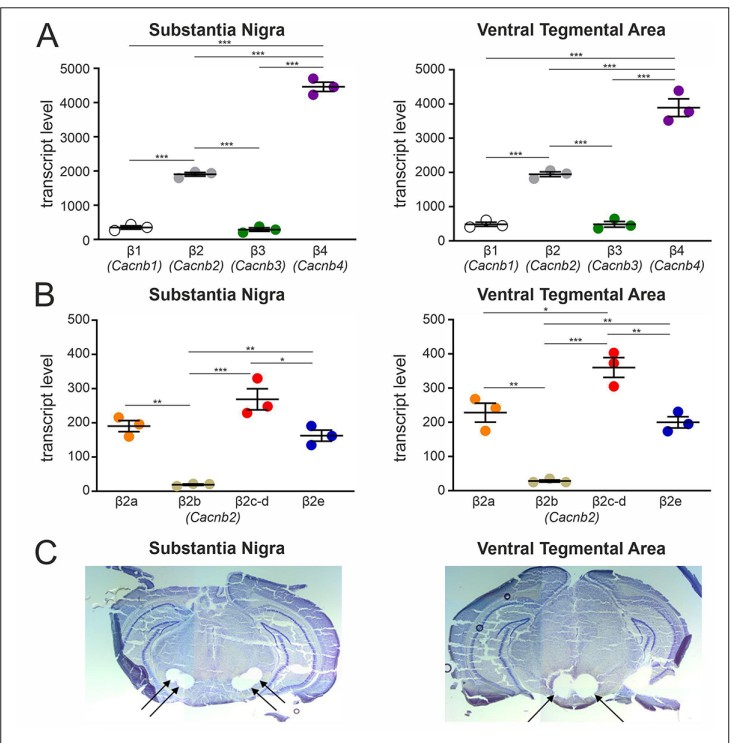

**Figure 2.** Transcript expression of various β-subunits and β2-subunit splice variants in mouse SN and VTA tissue. (**A**) Expression of β1-β4 subunit transcripts in SN (n=3, N=3) (left) and VTA (n=3, N=3) (right) determined by RT-qPCR as described in Methods. (**B**) Expression of β2a-β2e subunit transcripts in SN (n=3, N=3) (left) and VTA (n=3, N=3) (right). Data are shown as the mean ± SEM. Statistical significance was determined using one-way ANOVA followed by Bonferroni post-hoc test: *** $p < 0.001$; *p $P < 0.01$; * $p < 0.05$. Data was normalized to Gapdh and Tfrc determined by geNorm. (**C**) Example for four SN (left) and two VTA (right) tissue punches obtained for cDNA preparation with diameters of 0.5 mm each (left) or 0.8 mm each (right) from 7 to 8 successive 100-µm-sections between Bregma –3.00 mm and –3.80 mm, stained with Cresyl violet. Source data provided in *Figure 2—source data 1*.

The online version of this article includes the following source data and figure supplement(s) for figure 2:

**Source data 1.** Source data for data shown in *Figure 2*.

**Figure supplement 1.** Binding specificity of β1-β4 and β2a-β2e TaqMan assays and expression stability of endogenous control genes.

**Figure supplement 1—source data 1.** Source data for data shown in *Figure 2—figure supplement 1B and C*.

**Figure supplement 2.** Expression of β2-subunit splice variants in UV-LMD SN DA neurons.

**Figure supplement 2—source data 1.** Source data for data shown in *Figure 2—figure supplement 2A and B*.

Taken together, these data suggest that through their effect on inactivation gating β2a- and β2e-subunits could contribute to sustained Cav2.3 activity and AP-associated Ca$^{2+}$ transients previously recorded in SN DA neurons (*Benkert et al., 2019*).

## β-subunit transcripts in mouse SN and VTA

This prompted us to investigate if β2a and β2e splice variants are indeed expressed in SN tissue. First, we investigated β1 - β4 subunit expression patterns in the SN (and VTA for comparison, *Figure 2A*) dissected from brain slices of 12- to 14-week-old male C57Bl/6 N mice (*Figure 2C*) using a standard-curve-based absolute RT-qPCR assay (*Schlick et al., 2010*; *Figure 2—figure supplement 1*, *Supplementary file 1*, *Supplementary file 2*, *Supplementary file 3*). In both SN and VTA tissue, β4 (SN:~64%; VTA:~57%) and β2 (SN:~27%; VTA:~29%) represented the most abundant β-subunit transcripts, followed by β1 and β3 (~4–7%) (*Figure 2A*). Our findings are in excellent agreement (β2: 31–35%, β4: 41–45%) with cell-type-specific RNA sequencing data from identified mouse midbrain DA neurons (*Brichta et al., 2015*; *Shin, 2015*).

Since the relative abundance of β2-subunit splice variants in the brain is unknown, we used our standard curve-based RT-qPCR assay to specifically quantify their expression in the SN and VTA (*Figure 2B*; for alignment of N-terminal β2 splice variants see *Figure 2—figure supplement 1A*). Assays were designed to specifically discriminate between β2a, β2b, and β2e. β2c and β2d N-termini were detected together (the β2d N-terminus is also present in β2c but with different alternative splicing in the HOOK region; *Buraei and Yang, 2010*; see methods). In SN and VTA, β2a (~30%) and β2e (~26%) transcripts together comprised about half of all tested β2-subunit splice variants, cytosolic β2c and β2d-species about 42% and β2b only about 3% (*Figure 2B*). Therefore, β2a and β2e together should be able to form a substantial fraction of Cav2.3 channel complexes in these neurons.

We further confirmed the presence of the various N-terminal β2 splice variants in individual UV-laser microdissected mouse SN DA neurons at the mRNA level using a qualitative PCR approach (*Figure 2—figure supplement 2*, *Supplementary file 4*, *Supplementary file 5*). Moreover, quantitative RNAscope analysis confirmed the expression of β2e and β2a in identified mouse SN and VTA DA neurons with β2e more abundantly expressed compared to β2a (*Figure 2—figure supplement 2*).

## β2 splice variant-dependent regulation of Cav2.3 activity during SN DA neuron-like regular pacemaking activity

To test a possible role of β-subunits for Cav2.3 channel availability in SN DA neurons, we mimicked their electrical activity in tsA-201 cells because individual Ca$^{2+}$-current components arising from Cav2.3 channel complexes associated with different β-subunits or even different splice variants cannot be isolated during continuous pacemaking in patch-clamp recordings of identified SN DA neurons. With co-expressed β3 and β4 isoforms simulated SN DA neuron regular pacemaker activity (initiated from a holding potential of −89 mV) induced large inward currents in response to single AP waveforms (*Figure 3A and B*). Cav2.3 channels conducted I$_{Ca}$ during the repolarization phase of the AP (I$_{AP}$) without evidence for inward current during the interspike interval (ISI, *Figure 3A and B* bottom inset). However, I$_{AP}$ decreased rapidly during continuous activity and almost completely disappeared after 1 (co-transfected β3)–2 min (co-transfected β4; *Figure 3A, B, E and F*). The time course of I$_{AP}$ decrease was best fitted by a bi-exponential function (see legend to *Figure 3*). Our data, therefore, suggest that Cav2.3e α1-subunits, in complex with α2δ1 and β3 or β4, cannot support substantial inward currents during continuous SN DA neuron pacemaking activity. This is in contrast to our previously published finding of stable Cav1.3 Ca$^{2+}$-channel activity persisting under near identical experimental conditions (Cav1.3 α1/α2δ1/β3 previously published data; *Ortner et al., 2017*; illustrated for comparison in *Figure 3E and F*, in gray).

In contrast, the depolarizing shifts in steady-state inactivation and the slowing of inactivation kinetics by β2a and β2e subunits are sufficient to stabilize Cav2.3e currents during simulated regular pacemaking activity. When analyzed in parallel under identical experimental conditions I$_{AP}$ decreased with a much slower time course with about 40% of the maximal initial I$_{AP}$ still remaining even after 5 min of continuous activity (*Figure 3C–F*). The time course of I$_{AP}$ decrease could be fitted by a bi-exponential function predicting a steady-state reached at 32.3% ± 0.77% (n=12) of the initial I$_{AP}$ for β2a and 25.0% ± 0.12% (n=9) for β2e (see legend to *Figure 3*). Similar to co-transfected β3 or β4 subunits, β2a or β2e supported Cav2.3 I$_{Ca}$ predominantly during the repolarization phase after the AP

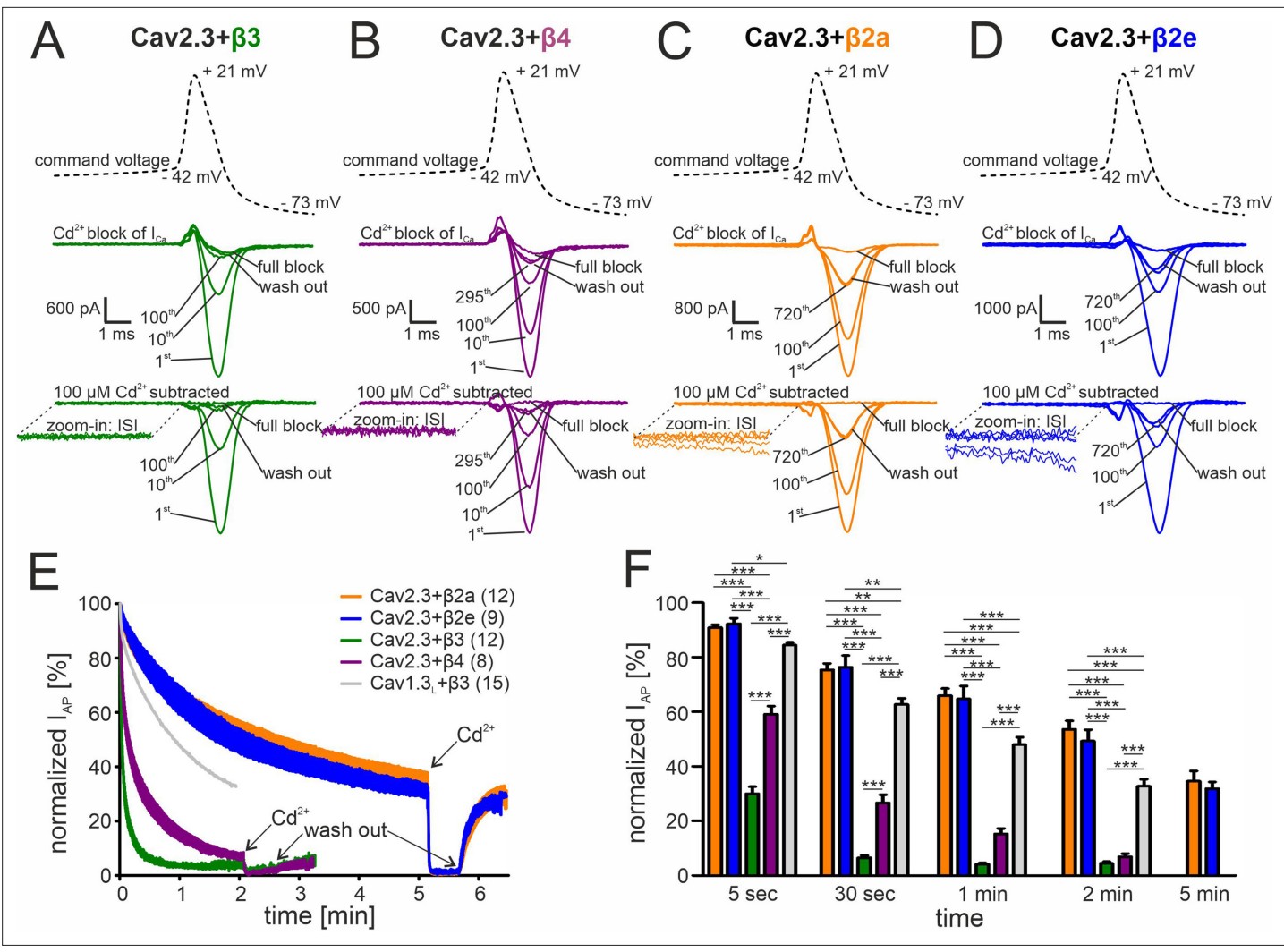

**Figure 3.** Activity-dependent inactivation of Cav2.3 channels co-transfected with different β subunits (and α2δ1) during simulated SN DA neuron regular pacemaking activity in tsA-201 cells. (**A–D**) Top panel: The SN DA neuron-derived command voltage was applied with a frequency of 2.5 Hz (only a time interval around the AP-spike is shown). Middle panel: Corresponding representative $Ca^{2+}$ current traces (2 mM charge carrier) for Cav2.3 channels co-expressed with α2δ1 and β3 (A, green), β4 (B, purple), β2a (C, orange), or β2e (D, blue) are shown for the indicated sweep number (1st, 10th, 100th, 295th, 720th). Cav2.3 currents were completely blocked by 100 µM Cadmium ($Cd^{2+}$), and remaining $Cd^{2+}$-insensitive current components were subtracted off-line (bottom panel). ISI, interspike interval. (**E**) Current decay during simulated 2.5 Hz SN DA neuron pacemaking. Normalized peak inward current during APs ($I_{AP}$) is plotted against time as mean ± SEM for the indicated number of experiments. $I_{AP}$ amplitudes were normalized to the $I_{AP}$ amplitude of the first AP after holding the cell at –89 mV. Cav1.3$_L$ co-expressed with α2δ1 and β3 (gray, mean only) is shown for comparison (data taken from ***Ortner et al., 2017***). The $I_{AP}$ decay was fitted to a bi-exponential function (Cav2.3 β3: $A_{slow}$ = 39.4% ± 0.65 %, $\tau_{slow}$ = 22.2 ± 0.15 min, $A_{fast}$ = 54.2 ± 0.76 %, $\tau_{fast}$ = 2.86 ± 0.07 min, non-inactivating=4.47% ± 0.13%; β4: $A_{slow}$ = 48.5 ± 0.26 %, $\tau_{slow}$ = 90.3 ± 1.07 min, $A_{fast}$ = 41.8 ± 0.40 %, $\tau_{fast}$ = 8.39 ± 0.16 min, non-inactivating=5.12% ± 0.12%; β2a: $A_{slow}$ = 52.6 ± 0.47 %, $\tau_{slow}$ = 299.3 ± 10.2 min, $A_{fast}$ = 13.4 ± 0.53 %, $\tau_{fast}$ = 18.2 ± 1.47 min, non-inactivating=32.3% ± 0.77%; β2e: $A_{slow}$ = 67.7 ± 0.11 %, $\tau_{slow}$ = 294.1 ± 1.77 min, $A_{fast}$ = 7.10.0±0.27 %, $\tau_{fast}$ = 16.6 ± 1.24 min, non-inactivating=25.0% ± 0.12%). Curves represent the means ± SEM for the indicated number of experiments (independent transfections, N=Cav2.3/β3: 4; Cav2.3/β4: 2; Cav2.3/β2a: 4; Cav2.3/β2e: 2; Cav1.3$_L$/β3: 6). (F) Normalized $I_{AP}$ decay after predefined time points for Cav2.3 with β3, β4, β2a or β2e, and Cav1.3$_L$ (with β3). Color-code and n-numbers as in panel E. Statistical significance was determined using one-way ANOVA followed by Bonferroni post-hoc test: *** p<0.001; ** p<0.01; * p<0.05. Source data provided in ***Figure 3—source data 1***.

The online version of this article includes the following source data for figure 3:

**Source data 1.** Source data for data shown in ***Figure 3***.

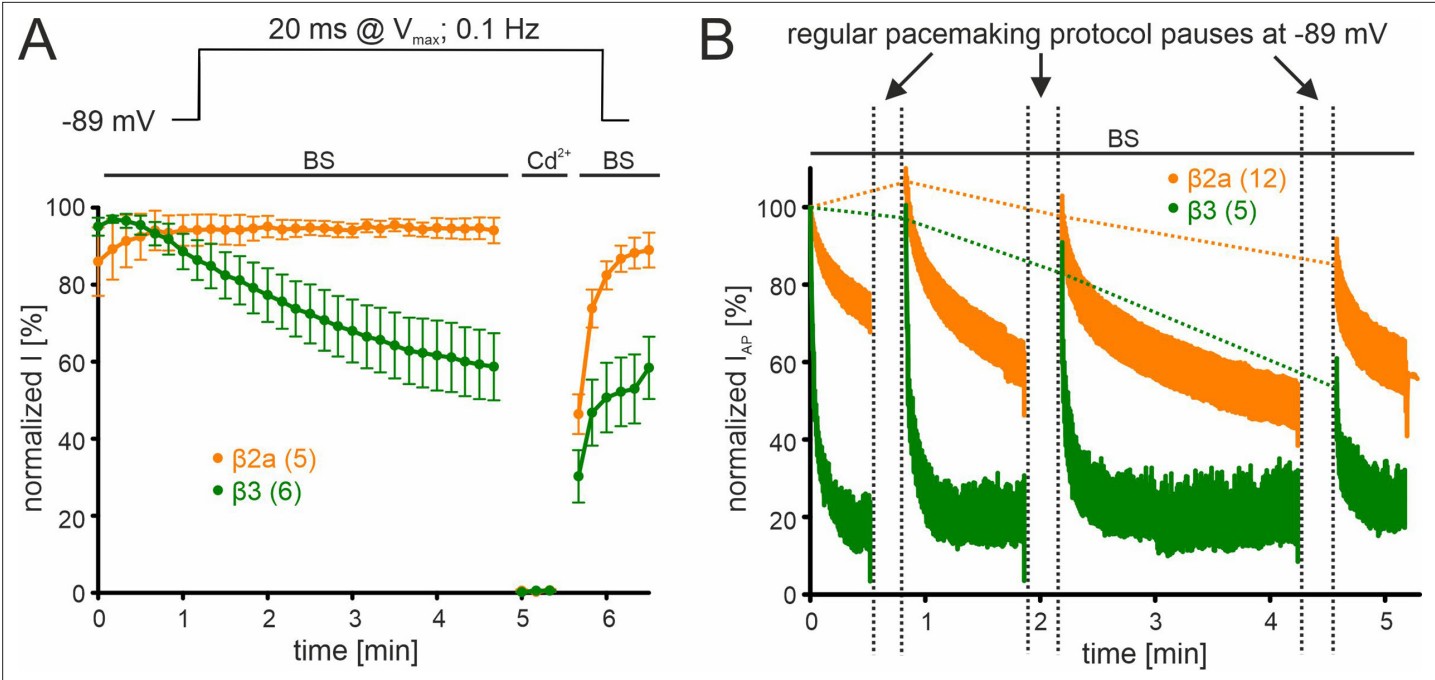

**Figure 4.** β-subunit-dependent run-down of Cav2.3 channel Ca$^{2+}$ current in tsA-201 cells. Data for Cav2.3 co-expressed with α2δ1 and β2a (orange) or β3 (green) are shown. (**A**). Run-down during a 0.1 Hz square pulse protocol (20 ms to V$_{max}$, holding potential –89 mV). Currents were normalized to the I$_{Ca}$ of the sweep with the maximal peak inward current observed during the recording. After a full block with 100 μM Cd$^{2+}$ currents recovered to the amplitude preceding the Cd$^{2+}$ application. (**B**) Cells were held at –89 mV and then stimulated using the regular SN DA neuron pacemaking protocol for 30 s, 1 min, and 2 min each followed by 20 s long pauses (vertical dashed lines) at hyperpolarized potentials (–89 mV) to allow channel recovery from inactivation. I$_{AP}$ of individual APs was normalized to the inward current of the first AP. The current run-down component can be estimated from the non-recovering current component (horizontal dashed lines). Traces represent means ± SEM from the indicated number of experiments (N=2). BS, bath solution.

spike. However, in accordance with enhanced window currents at negative voltages (*Figure 1D*) these subunits also supported an inward current during the interspike interval (ISI) as evident from the first few sweeps (with the largest current amplitudes) (*Figure 3C and D*, bottom insets). I$_{AP}$ persisting after 2 min (β3, β4) or 5 min (β2a, β2e) of pacemaking was completely blocked by 100 μM Cd$^{2+}$ with almost complete recovery upon washout (*Figure 3A–E*).

Irreversible loss of I$_{Ca}$, a phenomenon also known as current "run-down" widely described for both native and recombinant Cavs (*Kepplinger et al., 2000*; *Ortner et al., 2017*; *Schneider et al., 2018*) during patch-clamp recordings, may also contribute to the current decay observed during simulated pacemaking. We, therefore, quantified the contribution of current run-down for Cav2.3 co-expressed with α2δ1 and β2a or β3 subunits first by applying short (20 ms) square pulses to V$_{max}$ (holding potential –89 mV) with a frequency of 0.1 Hz (*Figure 4A*). With this protocol (short pulse, long inter-sweep interval, hyperpolarized holding potential) activity- and voltage-dependent inactivation of Cav2.3 channels should be minimized. While β3 co-transfected Cav2.3 channels showed a time-dependent current run-down to about 60% of the peak I$_{Ca}$ after 5 min, β2a prevented the current decline during this period (*Figure 4A*). To further quantify the current run-down during simulated pacemaking, we interrupted the pacemaking protocol after different time periods by 20 s long pauses at –89 mV to allow channel recovery from inactivation (*Figure 4B*). Thus, the percent run-down can be estimated from the non-recovering current component (*Figure 4B*, horizontal dashed lines). After 30 s of pacemaking the current amplitude during the first AP after the pause was similar to I$_{AP}$ during the initial AP with both β-subunits (β3: 96.9% ± 3.63%, 95% CI: 86.8–107.0, n=5, N=2; β2a: 107.3% ± 3.66%, 95% CI: 99.3–115.4, n=12, N=2, of the initial I$_{AP}$). After the pause that followed another 1 min of pacemaking, 83.6% ± 7.33% (95% CI: 63.2–103.9) of the initial current was still recovered with co-transfected β3 but after 2 more minutes of pacemaking recovery decreased to 54.7% ± 6.27% (95% CI: 34.7–74.7, *Figure 4B*; ~45% run-down after 4.5 min in total, n=5). This time course is in good agreement with

values obtained using the square-pulse protocol (*Figure 4A*). Again, run-down was largely prevented by co-expressed β2a-subunits (*Figure 4B*; ~16% run-down after 4.5 min in total, n=12).

These data clearly demonstrate that the $I_{AP}$ decrease in response to simulated SN DA neuron pacemaking (2.5 Hz) is almost completely due to the reversible accumulation of Cav2.3 channels in inactivated states, an effect partially prevented by β2a and β2e.

## Cav2.3 Ca²⁺ currents during simulated SN DA burst firing activity

In addition to regular pacemaking activity (in vitro) or irregular single spike mode (in vivo), burst firing with transient increases in intracellular Ca²⁺-load has been associated with neurodegeneration and selective neuronal vulnerability in Parkinson's disease (*Dragicevic et al., 2015*; *Schiemann et al., 2012*). Thus, we investigated to which extent Cav2.3 Ca²⁺-channels can contribute to Ca²⁺-entry during bursts and after post-burst hyperpolarizing pauses of SN DA neurons. After reaching steady-state $I_{AP}$ during simulated SN DA neuron pacemaking (β4: 1–2 min, β2a/β2e: 5–6 min, see also *Figure 3E and F*) we applied a computer modeled typical three-spike burst, followed by a 1.5 s long afterhyperpolarization-induced pause at more negative potentials as a command voltage as previously described (*Ortner et al., 2017*; *Figure 5A*). First, we quantified to which extent total burst Ca²⁺-charge, that is $I_{Ca}$ integrated over the duration of the burst, changes as compared to total Ca²⁺-charge during the same duration of a single AP in steady-state (calculated as the mean of 3 APs preceding the burst). Integrated $I_{Ca}$ during the burst was four- to sixfold higher than the mean integrated $I_{Ca}$ during a steady-state AP with all co-expressed β-subunits (β2a, β2e, β4) (*Figure 5B*, left panel). However, it has to be considered that with co-expressed β4 only ~6% of the initial $I_{AP}$ remained in steady-state during regular pacemaking (*Figure 3E and F*). Therefore, this relative increase will cause a much smaller rise in absolute Ca²⁺ charge compared to β2a/β2e-associated Cav2.3 where ~ 40% of $I_{AP}$ persisted in steady-state (*Figure 5B*, *Figure 3E and F*).

We also investigated if post-burst afterhyperpolarizations would allow Cav2.3 channels to recover from inactivation and thus mediate increased Ca²⁺-entry during the first APs after the burst. We first determined the recovery from inactivation of Cav2.3 channels co-expressed with α2δ1 and β4 or β2a using a square-pulse protocol (*Figure 5C*, *Table 2*; 1 s conditioning prepulse to $V_{max}$ to inactivate Cav2.3 channels followed by a 10 ms step to $V_{max}$ after different time periods at –89 mV). About 30% of currents recovered under these experimental conditions after 1.5 s at –89 mV with both co-expressed β4 and β2a (~sixfold increase of the remaining $I_{Ca}$ after 1 s at $V_{max}$). In contrast, recovery during the hyperpolarizing pause after the burst of the AP protocol was much less pronounced (β4) or absent (β2a, β2e; *Figure 5B*, right panel). This may be due to the different pulse protocols inducing channel inactivation over different time periods and may stabilize inactivated states with different recovery times.

Taken together, these data predict that β2a- and β2e-associated Cav2.3 channels can contribute to enhanced Ca²⁺-entry during brief burst activity but not during post-burst APs.

## Steady-state activation and inactivation of R-type currents in mouse DA neurons

We have recently shown in whole-cell patch-clamp recordings that 100 nM SNX-482 inhibit ~30% of total Cav currents in mouse SN DA neurons (*Benkert et al., 2019*) when stimulated from a negative holding potential of –70 mV. If $V_{0.5,inact}$ of Cav2.3 channels in SN DA neurons is indeed stabilized at more positive potentials, then the voltage-dependent inactivation of the R-type currents should allow a fraction of channels to be available even at voltages positive to –40 mV.

Using whole-cell patch-clamp recordings (*Figure 6*), we therefore measured the steady-state inactivation of R-type $I_{Ca}$ in identified SN DA neurons in mouse midbrain slices. First, R-type current (i.e. current reminaing after blocking other Cav channels) was isolated as reported in previous publications by preincubation of slices with selective inhibitors of Cav1 (1 µM ISR), Cav2 (Cav2.1, Cav2.2; 1 µM ω-conotoxin MVIIC) and Cav3 (10 µM Z941) (*Figure 6A*, blue traces/symbols, see also methods section). R-type current evoked from –100 mV activated at ~8 mV more positive voltages than total $I_{Ca}$ (*Figure 6A*, *Table 3* for statistics). This observation is in accordance with the positive activation voltage-range of Cav2.3e channels measured in tsA-201 cells. The R-type current $V_{0.5,inact}$ was ~5 mV more positive compared to total $I_{Ca}$ (–47.5 mV vs. –52.7 mV; *Figure 6A*, *Table 3*) and therefore ~7 mV more negative than recombinant Cav2.3 currents expressed with β2a or β2e subunits (~ –40 mV,

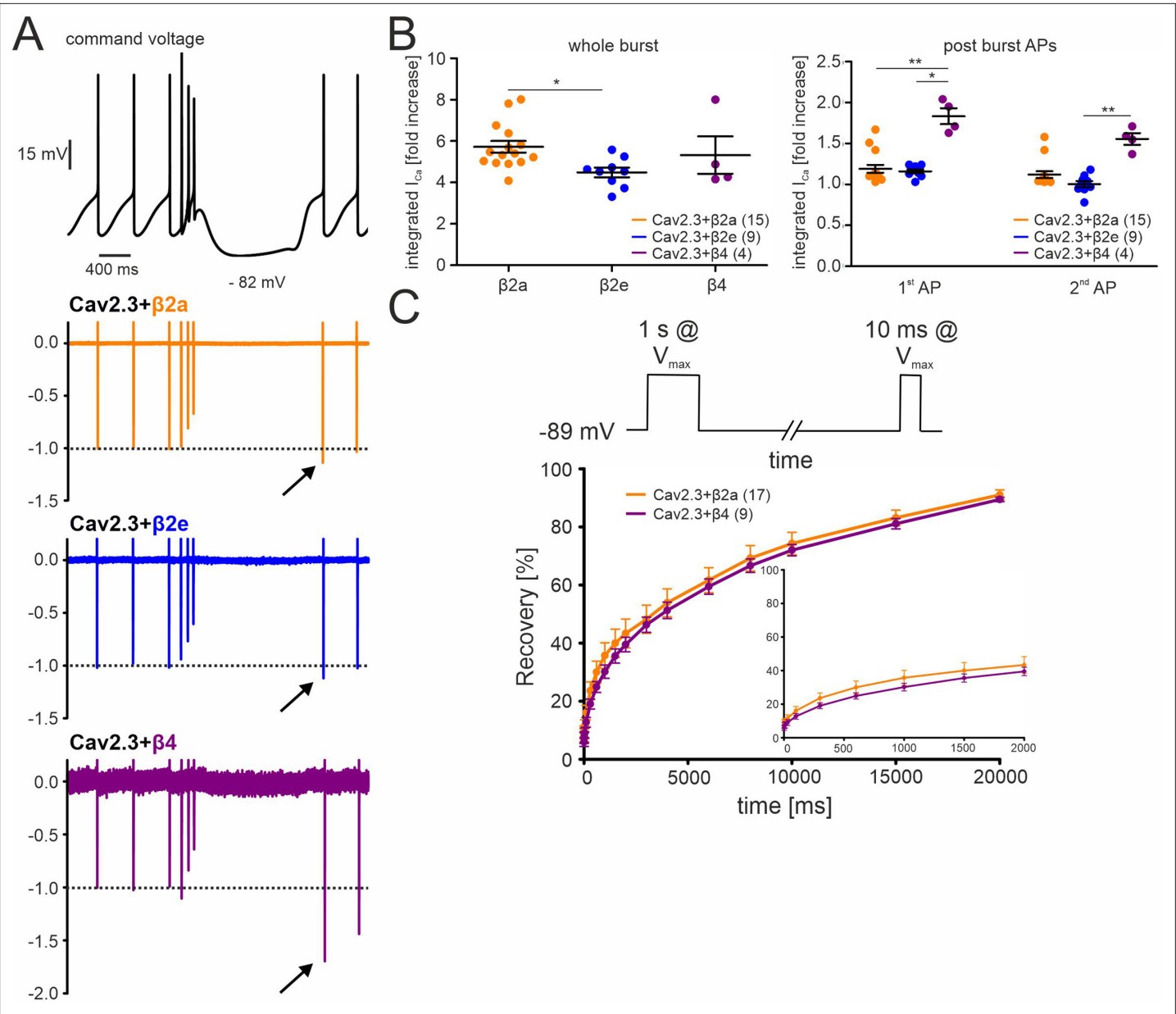

**Figure 5.** Effects of different β-subunits on Cav2.3 currents during a simulated SN DA neuron three-spike burst and post-burst APs in tsA-201 cells. The burst command voltage was elicited after ~5–6 min (β2a, β2e) or ~1–2 min (β4) of regular pacemaking to reach steady-state $I_{AP}$ (β2a and β2e:~30% of the initial $I_{AP}$, β4:~6% of the initial $I_{AP}$, see *Figure 3*). (**A**) Normalized current responses of Cav2.3 channels co-expressed with β2a, β2e or β4 subunits (and α2δ1) induced by a command voltage (top panel) simulating a typical three-spike burst followed by a hyperpolarization phase at hyperpolarized potentials (lowest voltage: –82 mV) for 1.5 seconds. Remaining $Cd^{2+}$-insensitive current components (100 µM $Cd^{2+}$) were subtracted off-line to extract pure Cav2.3 mediated $I_{Ca}$. One of at least four experiments with similar results is shown. (**B**) The integrated $I_{Ca}$ during a single AP before the burst (obtained as the mean of the three preceding APs) was set to 100% and compared with $I_{Ca}$ during the three-spike burst integrated over the time period equivalent to one AP (left) or the first APs after the pause (right). All investigated β-subunits resulted in increased integrated $I_{Ca}$ during the burst. Data represent the means ± SEM for the indicated number of experiments (N = β2a: 4; β2e: 2; β4: 2). Statistical significance was determined using one-way ANOVA followed by Bonferroni post-test (whole burst) or Kruskal-Wallis followed by Dunn's multiple comparison test (post-burst APs): *** p<0.001; ** p<0.01; * p<0.05. (**C**) Square-pulse protocol (top) used to determine recovery from inactivation after the indicated time intervals for β2a and β4-associated Cav2.3 channels (see Materials and methods for details). Data represent the means ± SEM for the indicated number of experiments (N=3). For statistics see *Table 2*. Source data provided in *Figure 5—source data 1*.

The online version of this article includes the following source data for figure 5:

**Source data 1.** Source data for data shown in *Figure 5*.

**Table 2.** Recovery from inactivation of Cav2.3 channels co-transfected with either β2a or β4 in combination with α2 δ 1 in tsA-201 cells.

All values are presented as the mean ± SEM for the indicated number of experiments (N=3). The r values represent the fraction of recovered $I_{Ca}$ after 100, 1500, 4000, or 10,000 ms at –89 mV between depolarizations to $V_{max}$. No statistical significance was observed (unpaired Student's t-test). Source data provided in *Table 1—source data 1*.

Cav2.3 - recovery from inactivation 2 mM $Ca^{2+}$

| β-subunit | r100 [%] | r1500 [%] | r4000 [%] | r10000 [%] | n |
|---|---|---|---|---|---|
| β2a | 15.7±2.4 | 39.3±4.6 | 53.4 ±4.5 | 74.2±3.6 | 17 |
| β4 | 12.7±1.7 | 35.6±2.4 | 51.3 ±2.8 | 72.0±1.9 | 9 |

The online version of this article includes the following source data for table 2:

**Source data 1.** Source data for data shown in *Table 2*.

*Table 1*) and ~23–29 mV more positive than recombinant Cav2.3 currents expressed with β4 or β3. Since R-type currents may include current components not mediated by Cav2.3 (*Newcomb et al., 1998*; *Sochivko et al., 2002*; *Tottene et al., 2000*; *Wilson et al., 2000*), we also isolated SNX-482-sensitive currents, which must be mediated by Cav2.3 (*Zamponi et al., 2015*) by subtracting current remaining in the presence of SNX-482 from control current (*Figure 6B*). Although the slope of the voltage-dependence of activation was different for the SNX-482-sensitive current as compared to R-type current, steady-state inactivation was again observed at significantly more positive voltages as compared to control current (–44.6 mV vs. –52.9 mV, *Figure 6B*, *Table 3*), and ~5 mV more negative than recombinant Cav2.3 currents associated with β4 or β3 (*Table 3*). Although these experiments do not directly prove the contribution of β2a and/or β2e to R-type current in SN DA neurons, the strong requirement of β-subunits for Cav2.3 channel activity and the inability of other cytosolic β-subunit variants to account for the observed voltage-dependence of R-type current (*Figure 1A and B*) strongly suggests a contribution of these membrane-associated β-subunits for preventing complete inactivation of Cav2.3-mediated currents during typical activity patterns in these cells.

We independently also tested the availability of SNX-482-sensitive R-type currents at more depolarized voltages using perforated patch recordings (*Figure 6—figure supplement 1*), which allow very stable recordings in identified SN DA neurons (*Figure 6—figure supplement 1I, J*). We held cells at a more positive holding potential of –60 mV, a voltage within the spontaneous depolarization phase of the ISI, where a substantial fraction of Cav2.3 channels must already be inactivated (*Figure 6*). Indeed, under these conditions still about 13% of the total $I_{Ca}$ was inhibited by acute bath application of SNX-482 and inhibition was partially reversible upon washout (*Figure 6—figure supplement 1A-D*, *Supplementary file 7*). Under these conditions, about 10% of current were blocked reversibly by the L-type channel blocker nifedipine (10 µM, *Figure 6—figure supplement 1E-H*).

To further confirm the presence of inactivation-resistant R-type currents, we also recorded SNX-482-sensitive $Ca^{2+}$ currents elicited from a holding potential of –70 mV in primary cultures of mouse DA midbrain neurons after 8–9 days in vitro. When added after the complete block of LTCC currents by 3 µM isradipine (comprising 24.7% ± 4.2% of total $Ca^{2+}$-current, n=15), 100 nM SNX-482 significantly reduced non-LTCC currents by 41% ± 4% (95% CI: 33–49, n=20, N=4, paired Student's t-test; $P<0.001$) (*Figure 7A–D*). All residual $I_{Ca}$ components were blocked by adding 2 µM $Cd^{2+}$ to the bath solution (*Figure 7A and B*).

We also determined the voltage-dependence of steady-state activation (*Figure 7E*) and inactivation (*Figure 7F*) before (non-L, control) and during application of 100 nM SNX-482 and calculated parameters for the SNX-482-sensitive R-type current by subtracting the current resistant to 100 nM SNX-482 from the control current (legend to *Figure 7* for details). SNX-482-sensitive currents (blue) activated with a half maximal activation voltage ($V_{0.5,act}$ –9.2±0.6 mV) slightly more positive than non-LTCC current ($V_{0.5,inact}$ = -12.5 ± 0.6 mV). To determine the steady-state inactivation, test pulses of 50 ms to 0 mV were preceded by 1 min prepulses to voltages from –80 to –10 mV (*Figure 7F*). SNX-482-sensitive current inactivation was best fit with a $V_{0.5,inact}$ of –58.1±0.6 mV, which was 12–18 mV more positive than recombinant Cav2.3 currents expressed with β4 or β3 (*Table 1*). Despite an ~13 mV

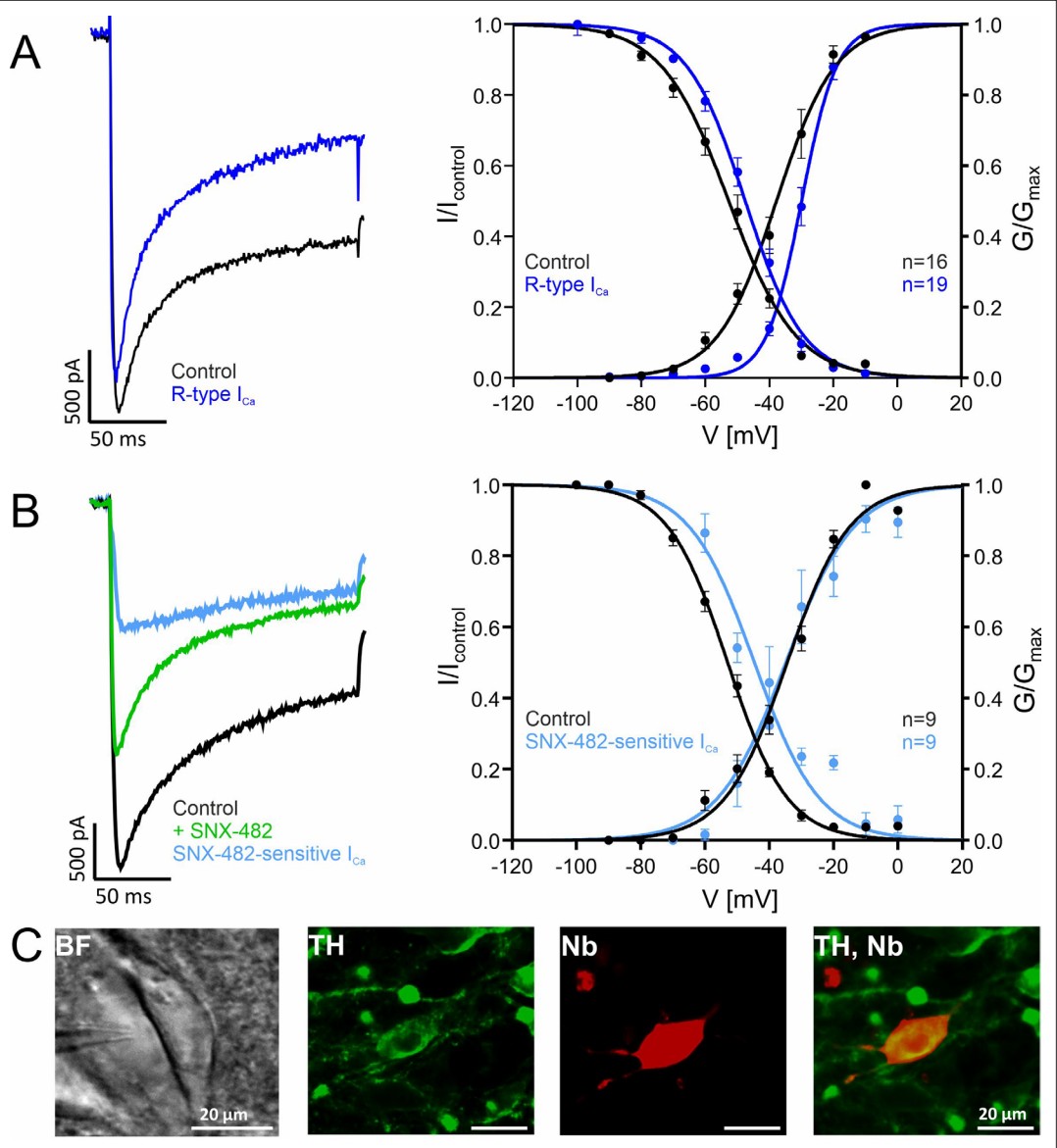

**Figure 6.** Voltage-dependence of gating of R-type currents in mouse SN DA neurons recorded in brain slices. (**A**) $I_{Ca}$ recorded in SN DA neurons without (Control, black, n=16) or after preincubation ("R-type", blue, n=19) of slices with a Cav channel blocker cocktail to inhibit Cav3 (10 µM Z941), Cav1 (1 µM I), Cav2.1 and Cav2.2 (1 µM $\omega$-conotoxin-MVIIC) (see Methods). Left panel: representative current traces of recordings at –20 mV test potentials (holding potential –100 mV); notice that similar amplitudes of control and R-type current were chosen to facilitate comparison of current kinetics. Right panel: voltage-dependence of steady-state activation and inactivation, curve fits to a Boltzman equation of all individual data points are shown. Data are means ± SEM. For parameters and statistics see **Table 3**. (**B**). SNX-482-sensitive currents (light blue) were obtained by subtracting current measured in the presence of SNX-482 (green) from the respective control $I_{Ca}$ before addition of SNX-482 (black) (n=9). Left panel: Representative recording of a SNX-482 sensitive current component compared to the respective control (holding potential –100 mV, –10 mV test potential). Steady-state activation and inactivation of control and SNX-482-sensitive current. Test protocols: holding potential –100 mV; voltage-dependence of activation: 150 ms test potentials to indicated voltages; voltage-dependence of inactivation: 5 s conditioning pulses to indicated voltages preceded ($I_{control}$) and followed by 20 ms test pulses to $V_{max}$. (**C**) Exemplary neuron as seen under the patch-clamp microscope with patch pipette next to it (left; BF, brightfield) and a neuron after histochemical staining for tyrosine hydroxylase (TH, green) and neurobiotin (Nb, red). Detailed parameters and statistics are given in **Table 3**. Source data provided in **Figure 6—source data 1**.

The online version of this article includes the following source data and figure supplement(s) for figure 6:

*Figure 6 continued on next page*

*Figure 6 continued*

**Source data 1.** Source data for data shown in *Figure 6*.

**Figure supplement 1.** Inhibition of $I_{Ca}$ in SN DA neurons of adult (12 weeks) mice by 100 nM SNX-482.

**Figure supplement 1—source data 1.** Source data for data shown in *Figure 6—figure supplement 1*, panels C, D and G, H.

more negative inactivation range compared to SN DA neurons (*Figure 6*, *Table 3*), this still allowed substantial channel availability at voltages positive to –50 mV (*Figure 7F*, in contrast to currents with co-expressed cytosolic β-subunits *Figure 1D*).

Finally, we tested if SNX-482 also affects the spontaneous AP firing of these cultured midbrain neurons. In current-clamp recordings 100 nM SNX-482 significantly reduced the spontaneous firing frequency from 4.1±0.8 Hz (control, n=10, N=3; 95% CI: 2.1–6.1) to 1.1±0.2 Hz (SNX-482, n=10, 95% CI: 0.2–2.1, p=0.0036, paired Student's t-test; *Figure 7—figure supplement 1A, B*), decreased the regularity of pacemaking and also caused changes in AP shape (*Figure 7—figure supplement 1A-C*). Cav2.3-mediated currents may therefore also contribute to the pacemaking cycle in cultured DA neurons.

## Discussion

Although the modulation of Cavs by β-subunits and the characteristic gating changes induced by N-terminally membrane-anchored β2-subunit splice variants have been well described in the literature, the physiological significance of these findings remained underexplored. Here, we provide strong evidence for an important role of β-subunit alternative splicing for Cav2.3 $Ca^{2+}$-channel signaling during continuous SN DA neuron-like regular pacemaking activity. We show that only membrane-bound β2a and β2e stabilize Cav2.3 channel complexes with voltage-dependent inactivation properties preventing complete inactivation during the on-average depolarized membrane potentials encountered during the pacemaking cycle. This cannot only explain our previous finding of a substantial contribution of SNX-482-sensitive R-type currents to activity-dependent somatic $Ca^{2+}$-oscillations in SN DA neurons in brain slices (*Benkert et al., 2019*) but also to pacemaking in cultured DA neurons. We also provide evidence for the expression of β2a and β2e subunit splice variants in highly vulnerable mouse SN DA and in more resistant VTA DA neurons. Together with our finding that R-type currents in SN DA neurons are available within a voltage-range more positive than expected for association

**Table 3.** Voltage-dependence of activation and inactivation of $Ca^{2+}$-currents in SN DA neurons.

Whole-cell voltage-clamp experiments to evoke Cav-currents were performed as described in Methods in SN DA neurons under control conditions (control A) or after preincubation with a Cav-blocker cocktail blocking all but R-type Cav channels (10 µM Z941, 1 µM isradipine, 1 µM $\omega$-conotoxin-MVIIC). In a second set of experiments, $I_{Ca}$ steady-state current activation and inactivation parameters of individual SN DA neurons were first recorded in the absence (control B) and again after 10 min perfusion with 100 nM SNX-482. SNX-482-sensitive current parameters were obtained by subtracting currents in the presence of SNX-482 from the respective individual currents before drug wash-in. Voltages were not corrected for liquid junction potential (–5 mV). Data are given as means ± SEM. N: number of mice; n: number of recorded neurons. Pooled data from all experiments were fitted to the Boltzman equation. Statistically signicant differences were determined by comparing fits using the extra sum of squares F-test (Prism 9.1): ***, p<0.001, difference vs Control A; +++, p<0.001 vs Control B.

| | Activation | | | Inactivation | | |
|---|---|---|---|---|---|---|
| | $V_{0.5,act}$ (95% CI) [mV] | $k_{act}$ (95% CI) [mV] | n/N | $V_{0.5, inact}$ (95% CI) [mV] | $k_{inact}$ (95% CI) [mV] | n/N |
| Control A (without Cav-blocker cocktail) | –37.8 (–39.3,–36.3) | 9.17 (7.90, 10.6) | 16/8 | –52.7 (–53.9,–51.5) | 10.5 (–11.6,–9.48) | 16/8 |
| +Cav blocker cocktail ("R-type") | –29.9 (–30.8,–29.1)*** | 5.49 (4.73, 6.30)*** | 19/5 | –47.5 (–48.7,–46.4)*** | 9.15 (–10.2,–8.16) | 19/5 |
| Control B (before SNX-482) | –34.4 (–35.7,–33.2) | 9.68 (8.59, 10.8) | 9/6 | –52.9 (–53.7,–52.1) | 9.30 (–10.9,–8.61) | 9/6 |
| SNX-482 - sensitive current | –34.9 (–37.7,–32.0) | 10.6 (8.26, 13.3) | 9/6 | –44.6 (–46.9,–42.2)+++ | 10.1 (–12.1,–8.24) | 9/6 |

The online version of this article includes the following source data for table 3:

**Source data 1.** Source data for data shown in *Table 3*.

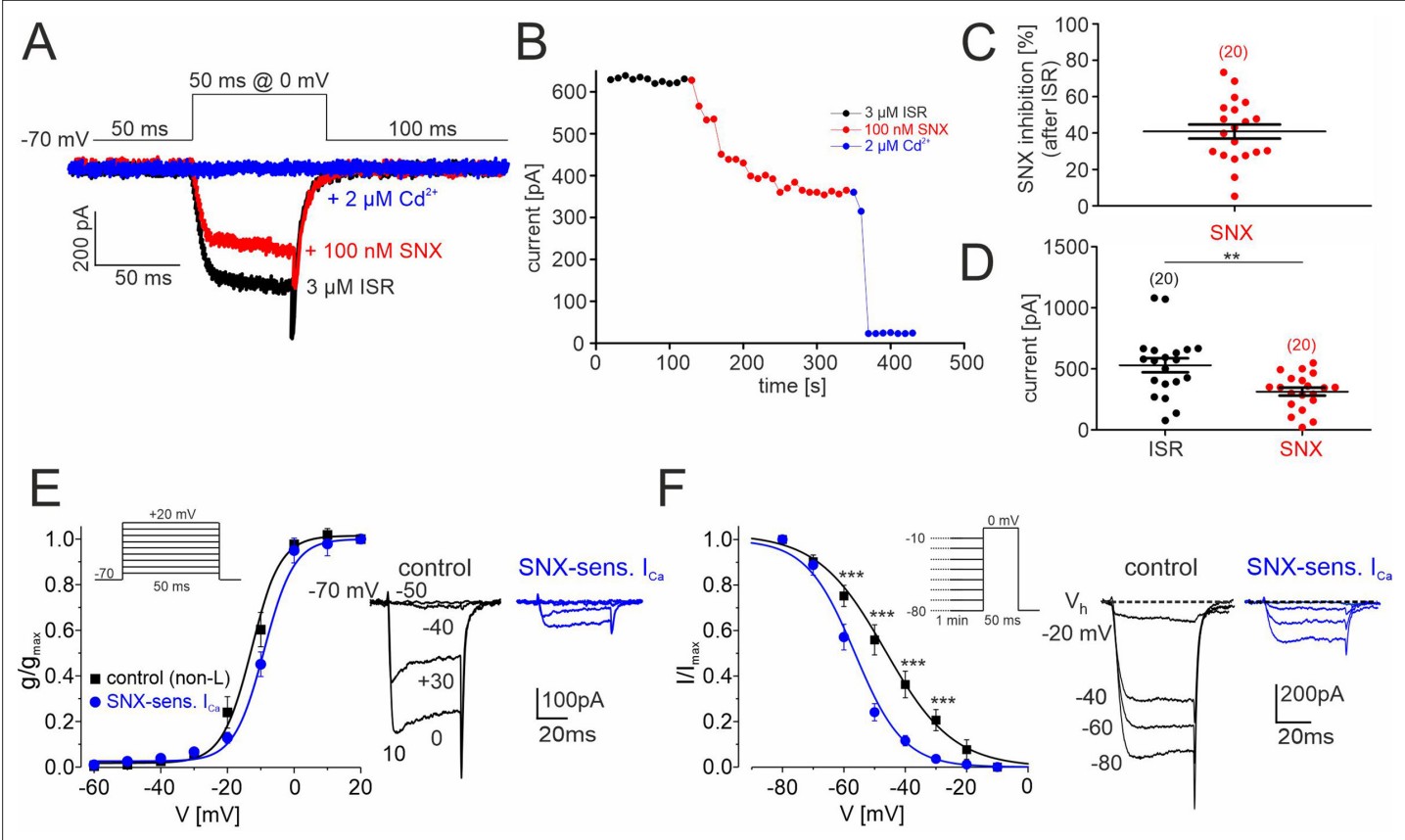

**Figure 7.** SNX-482–sensitive R-type currents in cultured mouse midbrain DA neurons. (**A**) Representative traces illustrating the inhibition of non-L-type $I_{Ca}$ by 100 nM SNX-482 (red). Cells were initially perfused with a bath solution containing 3 µM isradipine (black). Full block was obtained using 2 µM $Cd^{2+}$ (blue). Square pulses (50 ms) were applied to 0 mV from a holding potential of –70 mV (top) (**B**) Current amplitude values plotted as a function of time. After stabilization of $I_{Ca}$ with ISR (black circles), 100 nM SNX-482 was applied. The remaining currents was blocked by 2 µM $Cd^{2+}$. Current run-down in the absence of drugs during 150 s was less than 1% (0.48 ± 0.18%; n=4 cells). (**C**) SNX-482 inhibition expressed as % of control $I_{Ca}$ after LTCC block using 3 µM ISR. (**D**) Mean current amplitude at the end of ISR application and at the end of SNX-482 application. Absolute current amplitude decreased from 529±57 pA (95% CI: 409–649) to 313±33 pA (95% CI: 245–381, n=20, p<0.01, paired Students t-test). Data represent the means ± SEM for the indicated number of experiments (N=4). Statistical significance was determined using paired Student's t-test: *** p<0.001; **p<0.01; *p<0.05. (**E**) Left: Depolarizations of 50 ms were repeated every 10 s to the indicated potentials from a holding potential of –70 mV (inset). The voltage dependence of the conductance, g(V), was calculated with the equation $g(V)=I_{Ca}/(V- E_{Ca})$ with $E_{Ca}=+65$ mV for the currents recorded in the presence of 3 µM ISR (control; black squares; n=9–14 per voltage) and after subtraction of the currents insensitive to 100 nM SNX-482 (n=4–7) to yield (SNX-sensitive R-type; blue circles). Data were fitted to a Boltzman function. At each test potential (V) $I_{Ca}$ was estimated at the peak of the current trace. Right: representative current traces recorded during pulses to −50, –40, –30 and 0 mV. (**F**) Left: To determine the voltage-dependence of steady-state inactivation, test pulses of 50 ms to 0 mV were preceded by 1 min prepulses to voltages from –80 to –10 mV. This stimulation protocol (inset) was better tolerated by cultured DA midbrain neurons than the classical protocol used previously with other cells (*Calorio et al., 2019*; *Pinggera et al., 2015*). SNX-sensitive R-type currents (blue) were obtained after subtraction of the currents insensitive to 100 nM SNX-482 (n=3–8 per voltage) from non-L control current (black, n=8–15). Right: representative traces recorded at 0 mV from a holding potential of −20,–40, –60, and –80 mV. Inactivation curves could be best fit to Boltzman functions with the following parameters: R-type: $V_{0.5,inact}$ = -58.1±0.6 mV, k=−7.1 ± 1.2 mV; control: $V_{0.5,inact}$ = -48.0±0.9 mV, k=-11.1 ± 1.2 mV. The R-type currents (**E, F**) appear smaller than expected from the data in panels B and C (30–40% of non-L-type currents). This is due to a run-down of SNX-sensitive current during the repeated depolarizations used to determine the voltage-dependence of gating. This has only minimal effects on gating parameters. When accounting for a linear run-down of 40%, the $V_{0.5,act}$ would shift only from –58 to –56 mV. Source data provided in *Figure 7—source data 1*.

The online version of this article includes the following source data and figure supplement(s) for figure 7:

**Source data 1.** Source data for data shown in *Figure 7*.

**Figure supplement 1.** SNX-482 effects on pacemaking of cultured mouse midbrain DA neurons.

**Figure supplement 1—source data 1.** Source data for data shown in *Figure 7—figure supplement 1*, panel B.

with β4-, β3- or other cytosolic β2-subunits, our data strongly suggest that β2a and β2e contribute to non-inactivating Cav2.3 R-type currents required for normal DA neuron function. This may therefore also be of pathophysiological relevance given the previously reported pathogenic potential of Cav2.3 $Ca^{2+}$-channels in PD pathophysiology (**Benkert et al., 2019**).

The large shift of $V_{0.5,inact}$ of Cav2.3 R-type current by membrane-anchored β2a and β2e subunits would predict that Cav2.3 channel availability can be adjusted along a wide range of potentials (about 30 mV; $V_{0.5,inact}$ –40 to –70 mV, **Table 1**) just by varying the abundance and local availability of these subunits relative to cytosolic β-subunits within a cell or nanodomain. This range is also reflected by an ~11–14 mV difference in $V_{0.5,inact}$ of R-type currents in our recordings from mouse midbrain DA neurons in culture and ex vivo slice recordings from SN DA neurons. However, the voltage-range of steady-state inactivation in both preparations is clearly more positive than predicted from heterologous expression with β4- or β3-subunits. This could be due to a different contribution of β2a and/or β2e to Cav2.3 channel complexes in these preparations. Our data predict that this contribution is higher in SN DA neurons than in cultured cells.

Our quantitative PCR analysis of β-subunit expression demonstrate that, although β4 (60%) is the predominant one in the SN and VTA, β2-subunit transcripts are also abundant and comprise about 27% of all β-subunits. These data are in excellent agreement with RNAseq data from mouse midbrain DA neurons with expression levels of about 33% for β2% and 43% for β4 (**Brichta et al., 2015**; **Shin, 2015**) and transcriptomic analysis of laser-captured SN DA neurons from 18 human postmortem brains (30% β2, 45% β4; 15% β3, 8% β1; **Aguila et al., 2021**). While these transcriptomic data do not provide quantitative information about the expression of β2-splice variants, we clearly demonstrated the presence of both membrane-bound isoforms in identified SN DA neurons using PCR and RNAscope analysis.

Our transcript data (and confirmatory RNAseq data) show that β2a and β2e are present in SN DA neurons but still represent only a small fraction of all β-subunits present. However, our experimental data find a positive inactivation voltage-range suggesting that a majority of SNX-482-sensitive channels is associated with these β-subunits. At present biochemical data regarding preferential targeting of β-subunits to specific subcellular neuronal compartments and/or Cav-subtypes is absent but cannot be excluded. For example, β4-subunits may extensively target to channels at axonal presynaptic locations (as reported in cultured hippocampal neurons) or even to the nucleus (where they regulate gene transcription) and thus contribute less to Cav2.3 channel complexes at somatodendritic sites (**Etemad et al., 2014**). Moreover, differences in the affinity of β-subunits for their α1-subunit interaction domain in the I-II-linker may also support preferential association with β2 (**De Waard et al., 1995**).

Based on the finding that β-subunit association is required for expression of robust Cav2.3 channel currents (**Figure 1A**, **Table 1**), they must also play a key role in determining the gating properties of individual channel complexes. However, this does not exclude a modulatory role of other posttranslational modifications of Cav2.3 channels or protein interaction partners expressed at somatodendritic locations of SN DA neurons that could enhance Cav2.3 availability at more positive membrane potentials (such as Rab3-interacting proteins at axonal sites, **Kiyonaka et al., 2007**; **Robinson et al., 2019**).

A limitation of our work is that we cannot provide direct proof for a role of β2-subunit splice variants for R-type current modulation in DA neurons. Direct proof for the existence of such complexes would require biochemical studies requiring immunoprecipitation of Cav2.3 complexes from laser-dissected SN DA neurons. However, such studies, as well as other approaches, such as proximity ligation assays are hampered by the fact, that suitable β2-splice variant-specific antibodies are not available. We also found no suitable combination of antibodies raised in different species for such studies. Two mouse anti-β2 (not even splice variant-specific) antibodies that could be combined with our previously used rabbit anti-Cav2.3 α1 subunit antibody turned out to bind to other nonspecific bands or cross-reacted with β3 (not shown, data available upon request). Alternatively, direct proof for a role of β2-subunit splice variants could be obtained by a splice variant-specific gene knockout or siRNA-mediated knock-down of both β2a and β2e subunits in SN DA neurons, followed by isolation of Cav2.3 current components or $Ca^{2+}$ transients of which only a fraction would be mediated by β2a/β2e-associated channels. This would be methodologically extremely challenging in these neurons.

We have previously shown that Cav2.3-knockout mice are protected from the selective loss of SN DA neurons in the chronic MPTP/probenecid PD model (**Benkert et al., 2019**). At least in this rodent model of PD, the observed protection provides strong evidence for a role of these channels

in PD pathology. Pharmacological inhibition of Cav2.3 alone or together with other Cavs may therefore confer beneficial disease-modifying effects and a novel approach for neuroprotection in PD. The recent failure of the LTCC blocker ISR in the STEADY-PD III trial (*Investigators, 2020*) to prevent disease progression in early PD patients indicates that inhibition of LTCCs alone may not be sufficient for neuroprotection. Our previous preclinical findings identified also Cav2.3 channels as novel drug targets for neuroprotective PD-therapy. Therefore, the inhibition of Cav2.3 in addition to Cav1.3 may be required for clinically meaningful neuroprotection. However, Cav2.3-mediated R-type currents are notoriously drug-resistant (*Schneider et al., 2013*), and so far no selective potent small-molecule Cav2.3/R-type blocker has been reported.

Our findings reported here provide the rationale for exploring novel neuroprotective strategies based on Cav2.3 channel inhibition. These could take advantage of the predicted strong dependence of continuous Cav2.3 channel activity on gating properties such as those stabilized by β2a and/or β2e. Rather than inhibiting $Ca^{2+}$-entry through the pore-forming α1-subunit, such a strategy could aim at reducing its contribution to the R-type current component persisting during continuous activity in SN DA neurons by interfering with the association of β2a and β2e subunits. Even if other β-subunits would replace them in the channel complex and ensure its stable expression, our data suggest that they would not be able to shift the steady-state inactivation voltage into the operating voltage-range of SN DA neurons like β2a and β2e. Such an approach appears realistic due to the availability of novel genetically encoded $Ca^{2+}$-channel inhibitors for a cell-type-specific gene therapeutic intervention. One such approach (CaVablator) has elegantly been used to specifically target $Ca^{2+}$-channel β-subunits for degradation by fusing β-specific nanobodies with the catalytic HECT domain of Nedd4L, an E3 ubiquitin ligase (*Morgenstern et al., 2019*). This strongly reduces Cav1- and Cav2-mediated $Ca^{2+}$-currents in different cell types. At present, it is unclear to which extent other high-voltage-activated Cav2 channels (Cav2.1, Cav2.2) also contribute to the high vulnerability of SN DA neurons. However, our recent quantitative RNAscope analyses in mature SN DA neurons (*Benkert et al., 2019*) clearly demonstrate that Cav2.3 α1-subunit (*Cacna1e*) transcripts are the most abundant α1-subunit expressed in these cells, in excellent agreement with cell-type-specific RNAseq data of identified midbrain DA neurons (*Brichta et al., 2015*; *Shin, 2015*).

We also show that in cultured mouse DA neurons, 100 nM SNX-482 slows pacemaking. While SNX-482 is highly suitable to isolate Cav2.3-mediated $Ca^{2+}$ current components, effects on pacemaking also must take into account that this toxin is also a potent blocker of Kv4.3 channels (*Kimm and Bean, 2014*) underlying A-type $K^+$-currents ($I_A$). 60 nM nearly fully block reconstituted Kv4.3 currents in HEK cells ($IC_{50}$ ~3 nM). Therefore, one can argue that in current-clamp recordings, 50–300 nM SNX-482 could alter pacemaking or the AP shape by effectively blocking Kv4.3 channels. However, the observed SNX-482-induced shortening of APs and the reduction of spontaneous firing are difficult to reconcile with a block of $I_A$ channels, which typically induces a broadening of APs in several neuronal preparations (*Kim et al., 2005*) and an increased frequency in DA neurons (*Liss et al., 2001*).

Therefore, the observed SNX-482 effects in cultured neurons provided no final answer to the question about a potential role of Cav2.3 on pacemaking. In brain slices of Cav2.3 knockout mice, small changes in AP spike waveforms were observed, without major changes in pacemaking activity (*Benkert et al., 2019*). Similarly, inconsistent findings for the contribution of LTCC currents for pacemaking in slice recordings (*Dragicevic et al., 2014*; *Guzman et al., 2009*; *Mercuri et al., 1994*; *Ortner et al., 2017*) and cultured neurons (*Puopolo et al., 2007*) have been reported but the reasons for this remain unclear.

Despite providing no direct biochemical evidence, we here show the first example for a potential physiological and perhaps even pathophysiological role of β2-subunit alternative splicing emphasizing a need for further investigation in other types of neurons.

# Materials and methods

## Key resources table

| Reagent type (species) or resource | Designation | Source or reference | Identifiers | Additional information |
|---|---|---|---|---|
| Mouse strain:C57Bl/6N, C57Bl/6J | C57Bl/6N, C57Bl/6J | Charles River | C57Bl/6N, C57Bl/6J | |

*Continued on next page*

*Continued*

| Reagent type (species) or resource | Designation | Source or reference | Identifiers | Additional information |
|---|---|---|---|---|
| Mouse strain:C57Bl/6-TH-GFP mice | C57Bl/6-TH-GFP | *Matsushita et al., 2002*; *Sawamoto et al., 2001* | | GFP-expression under TH-promotor |
| Transfected construct | Human Cav1.3$_L$ α1-subunit (C-terminally long Cav1.3 splice variant) | *Koschak et al., 2001* | GenBank accession number EU363339 | Cloned into proprietary vector lacking GFP ("GFP$^{minus}$"), described in *Grabner et al., 1998* |
| Transfected construct | Human Cav2.3 α1-subunit | *Pereverzev et al., 2002* | | Cloned into pcDNA3 vector |
| Transfected construct | Rat β3-subunit | *Koschak et al., 2001* | GenBank: NM_012828, | Cloned into pCMV6 vector |
| Transfected construct | Rat β2a-subunit | *Koschak et al., 2001* | GenBank: M80545 | Cloned into pCMV6 vector |
| Transfected construct | Mouse β2d-subunit | β2aN1, *Link et al., 2009* | GenBank: FM872408.1 | Cloned into pCAGGS vector |
| Transfected construct | Mouse β2e-subunit | β2aN5, *Link et al., 2009* | GenBank: FM872407 | Cloned into pCAGGS vector |
| Transfected construct | $_{C3S/C4S}$β2a | Cysteine residues in position 3 and 4 of β2a replaced by serines, *Gebhart et al., 2010* | | Cloned into pCMV6 vector |
| Transfected construct | Rat β4-subunit (splice variant β4e) | *Etemad et al., 2014* | | Cloned into pβA vector |
| Cell line (*Homo sapiens*) | TsA-201 cells | European Collection of Authenticated Cell Cultures | ECACC, catalogue # 96121229, lot # 13D034 | Tested Mycoplasma negative |
| Antibody | Anti-tyrosine hydroxylase (TH; rabbit polyclonal) | Merck Millipore | Cat#: 657012, | 1:1000 |
| Commercial assay or kit | RNeasy Lipid Tissue Mini Kit | Qiagen GmbH, Germany | catalog # 1023539 | |
| Commercial assay or kit | Phenol/guanidine-based Qiazol lysis reagent | Qiagen GmbH, Germany, | catalog # 79,306 | |
| Commercial assay or kit | Maxima H Minus First Strand cDNA synthesis kit | Thermo Fisher Scientific, Waltham, MA, USA | catalog # K1682 | |
| Commercial assay or kit | Quant-IT PicoGreen dsDNA Assay Kit | Thermo Fisher Scientific, Waltham, MA, USA | catalog # P7589 | |
| Commercial assay or kit | Mycoplasma Detection Kit | ATCC | Universal Mycoplasma Detection Kit 30–1012 K | |
| Commercial assay or kit | TaqMan Universal PCR Master Mix | Thermo Fisher Scientific, Waltham, MA, USA | | |
| Commercial assay or kit | RNAscope technology | Advanced Cell Diagnostics (ACD) | RNAscope Fluorescent Multiplex Detection Kit (ACD, Cat# 320851) | |
| Chemical compound | Neurobiotin-labeling reagent: Streptavidin Alexa Fluor conjugate 647 | Thermo Fisher Scientific, Waltham, MA, USA | catalog # S21374 | 1:1000 |
| Chemical compound | SNX-482 | Alomone Labs, Jerusalem, Israel | catalog # RTS-500 | |
| Software, algorithm | Clampfit | Clampfit | Version 10.0 or 10.7, RRID:SCR_011323 | |
| Software, algorithm | QuantStudioTM | Thermo Fisher Scientific, Waltham, MA, USA | Quantstudio 3 Software | |
| Software, algorithm | Fiji | https://imagej.net/Fiji, *Schindelin et al., 2012* | | |
| Software, algorithm | GraphPad Prism | GraphPad | GraphPad Prism Ver 5, 7.04 and 8 (RRID:SCR_002798) | |
| Software, algorithm | SigmaPlot | Systat Software, Germany | SigmaPlot Ver 14, RRID:SCR_003210 | |

## Animals

For quantitative real-time PCR (RT-qPCR) experiments, male C57Bl/6 N mice were bred in the animal facility of the Centre for Chemistry and Biomedicine (CCB) of the University of Innsbruck (approved by the Austrian Animal Experimentation Ethics Board). For electrophysiological experiments of cultured midbrain DA neurons, C57Bl/6 TH-GFP mice (*Matsushita et al., 2002*; *Sawamoto et al., 2001*) were kept heterozygous via breeding them with C57Bl/6 mice (in accordance with the European Community's Council Directive 2010/63/UE and approved by the Italian Ministry of Health and the Local Organism responsible for animal welfare at the University of Torino; authorization DGSAF 0011710 P-26/07/2017). All animals were housed under a 12 hr light/dark cycle with food and water ad libitum. For whole cell voltage-clamp recordings of SN DA neurons in acute brain slices, as well as single-cell RT-qPCR, juvenile male C57Bl/6J mice (PN11-13) were bred at the animal facility of Ulm

University. For RNAscope analysis, adult male C57Bl/6J mice and Cav2.3 WT mice were bred at the animal facility of Ulm University. Animal procedures at the Universities of Ulm (Regierungspräsidium Tübingen, Ref: 35/9185.81–3; Reg. Nr. o.147) and Cologne (LANUV NRW, Recklinghausen, Germany (84–02.05.20.12.254) were approved by the local authorities).

## RNA isolation and cDNA synthesis for tissue RT-qPCR

Tissue was dissected after mice had been sacrificed by cervical dislocation under isoflurane (Vetflurane, Vibac UK, 1000 mg/g) anesthesia. The freshly extracted mouse brains from 12 to 14 weeks old male mice were snap-frozen in isopentane (Carl Roth, catalog #3926.2) that was pre-cooled with dry ice (–40 °C). To dissect VTA and SN tissue, 100-µm-thick sections were cut on a cryostat (CM1950, Leica, Germany) and collected on glass coverslips. After cooling the coverslips on dry ice, regions of interest were punched under a dissection microscope using a pre-cooled sample corer (Fine Science Tools, Germany) (VTA: inner diameter 0.8 mm, 1 punch per hemisphere; SN: inner diameter 0.5 mm, 2 punches per hemisphere). For each brain region, tissue punches from both hemispheres of 7–8 successive 100 µm sections between Bregma –3.00 mm and –3.80 mm (according to *Paxinos and Franklin, 2004*) were collected in the sample corer. The tissue punches were transferred into an Eppendorf tube and again snap-frozen in liquid nitrogen. The punched brain sections were stained with cresyl violet (Nissl's staining) for histological verification. Briefly, the slides with the punched sections were incubated in 4% PFA overnight. On the next day, the sections were rinsed in $H_2O$ and the following staining steps were performed: $H_2O$ 1 min, 70% ethanol 5 min, 100% EtOH 5 min, 0.5% cresyl violet solution pH 3.9 (2.5 g cresyl violet acetate, 30 ml 1 M Na-acetate*3 $H_2O$ and 170 ml 1 M acetic acid) 10 min, 70% ethanol 2 min, 100% ethanol 2 min, Xylol 1 min. The sections were coverslipped immediately with Eukitt (Sigma-Aldrich, catalog #03989), left to harden overnight and imaged on a bright-field microscope at 2 x.

RNeasy Lipid Tissue Mini Kit (Qiagen GmbH, Germany, catalog #1023539) was used to isolate total RNA from brain tissues. Briefly, the tissue was disrupted and homogenized by vortexing samples for 5–10 min in 500 µl (SN, VTA) of phenol/guanidine-based Qiazol lysis reagent (Qiagen GmbH, Germany, catalog #79306) and passing the lysate ten times through a 21-gauge needle. All further steps, including purification of RNA with QIAGEN silica gel membrane technology (Qiagen GmbH, Germany), were performed according to the manufacturer's protocol. For the final elution 15 µl of RNase-free water were used twice. An optional on-column DNase digestion (Qiagen GmbH, Germany, catalog #79254) was performed to reduce genomic DNA contamination. The RNA concentration was determined photometrically yielding approximately 20 ng/µl for VTA and SN or 1–2 µg/µl for whole brain.

RNA was reverse transcribed using Maxima H Minus First Strand cDNA synthesis kit with random hexamer primers following the manufacturer's instructions (Thermo Fisher Scientific, Waltham, MA, USA). One µg or 13 µl of total RNA were used as template for reverse transcription. One µl cDNA corresponds to 0.65 x the amount of RNA equivalent.

## Standard curve method-based RT-qPCR for quantification of β-subunit expression in SN and VTA

In order to generate DNA templates of known concentrations for RT-qPCR standard curves, the concentration of the digested fragments was determined using the Quant-IT PicoGreen dsDNA Assay Kit (Invitrogen, Carlsbad, CA, USA). Subsequently, standard curves were generated using a serial dilution ranging from $10^7$ to $10^1$ DNA molecules in water containing 1 µg/ml of poly-dC DNA (Midland Certified Reagent Company Inc, Midland, TX, USA). RT-qPCRs of standard curves and samples were performed as described previously (*Ortner et al., 2020*; *Schlick et al., 2010*). Samples for RT-qPCR quantification (50 cycles) contained 5 ng of total RNA equivalent of cDNA, the respective TaqMan gene expression assay, and TaqMan Universal PCR Master Mix (Thermo Fisher Scientific). Specificity of all assays was confirmed using different DNA ratios of corresponding and mismatched DNA fragments for the β1–4 and for the β2a-2e assays, respectively. Importantly, all assays specifically recognized the corresponding fragment even in the presence of a 10-fold higher concentration of other splice variants (*Figure 2—figure supplement 1*). RT-qPCR was performed in duplicates from three independent RNA preparations from three biological replicates. Samples without template served as negative controls. No RT RNA controls could not be included since all obtained RNA was transcribed into cDNA due

to low yields. The expression of seven different endogenous control genes was routinely measured and used for data normalization as previously described (*Vandesompele et al., 2002*; *Figure 2—figure supplement 1*, *Supplementary file 1*): *Actb* (β-actin), *B2m* (β2-microglobulin), *Gapdh* (Glyceraldehyde 3-phosphate dehydrogenase), *Hprt1* (Hypoxanthine phosphoribosyltransferase 1), *Tbp* (TATA box-binding protein), *Tfrc* (Transferrin receptor), and *Sdha* (Succinate dehydrogenase subunit A) (*Ortner et al., 2020*; *Schlick et al., 2010*). Briefly, data were normalized to the most stably expressed endogenous control genes (*Gapdh* and *Tfrc*) determined by geNorm (*Figure 2—figure supplement 1*, *Supplementary file 1*). Normalized molecule numbers were calculated for each assay based on their respective standard curve. Standard curve parameters are given in (*Supplementary file 3*). qPCR analyses were performed using the 7,500 Fast System (Applied Biosystems, Foster Systems, CA, USA).

## Quantitative RT-qPCR of mouse brain tissue samples

Fragments of β-subunits and β2 splice variants were amplified from mouse whole brain cDNA utilizing specific primers (*Supplementary file 2*) and subcloned into the Cav1.3 8 a 42 pGFP^minus vector after restriction enzyme digestion using SalI and HindIII. Primer sequences for β1-β4 have been described previously (*Schlick et al., 2010*), but additional SalI and HindIII restriction enzyme sites (underlined in *Supplementary file 2*) were inserted to allow subsequent ligation of fragments into the digested vector. TaqMan gene expression assays (Thermo Fisher Scientific, Waltham, MA, USA) and custom-made TaqMan gene expression assays were designed to span exon-exon boundaries (*Supplementary file 2*) as already described (*Ortner et al., 2020*).

The expression of β1, β2, β3, β4, β2a, β2b, β2c-d, and β2e was assessed using a standard curve method-based on PCR fragments of known concentration (*Ortner et al., 2020*; *Schlick et al., 2010*). β2c and β2d were detected by a common assay as selective primer design failed due to high sequence similarity. This assay binds at the exon-exon boundary of exons 2 A and 3 of β2c and β2d and also recognizes a number of splice variants comprising the β2d N-terminus but with different alternative splicing in the HOOK region of the subunit (*Buraei and Yang, 2010*). Details about assay specificity are given in *Figure 2—figure supplement 1* and *Supplementary file 3*.

## cDNA constructs

For transient transfections hCav2.3e (cloned into pcDNA3, *Pereverzev et al., 2002*) or hCav1.3$_L$ (human C-terminally long Cav1.3 splice variant; GenBank accession number EU363339) α1 subunits were transfected together with the previously described accessory subunit constructs: β3 (rat, NM_012828, *Koschak et al., 2001*), β2a (rat, M80545, *Koschak et al., 2001*), β2d (mouse, β2aN1, FM872408.1; *Link et al., 2009*), β2e (mouse, β2aN5, FM872407; *Link et al., 2009*, where β2d and β2e were kindly provided by V. Flockerzi, Saarland University, Homburg), $_{C3S/C4S}$β2a (cysteine residues in position 3 and 4 of β2a replaced by serines, *Gebhart et al., 2010*) or β4 (rat, splice variant β4e, kindly provided by Dr. Bernhard Flucher, Medical University Innsbruck; *Etemad et al., 2014*) and α2δ1 (rabbit, NM_001082276, *Koschak et al., 2001*).

## Cell culture and transfection

TsA-201 cells were obtained from the European Collection of Authenticated Cell Cultures (ECACC, catalogue number 96121229, lot number 13D034) at passage 6. Cell stocks of passage 8 were frozen and cultures were re-expanded from stocks for not more than 20 passages. Cell cultures were tested negative (Universal Mycoplasma Detection Kit 30–1012 K, American Type Culture Collection) for mycoplasma infection. TsA-201 cells were cultured as described (*Ortner et al., 2020*) in Dulbecco's modified Eagle's medium (DMEM; Sigma-Aldrich, catalog #D6546) that was supplemented with 10% fetal bovine serum (FBS, Gibco, catalog #10270–106), 2 mM L-glutamine (Gibco, catalog #25030–032), penicillin (10 units/ml, Sigma, P-3032) and streptomycin (10 µg/ml, Sigma, S-6501). Cells were maintained at 37 °C and 5% $CO_2$ in a humidified incubator and were subjected to a splitting procedure after reaching ~80% confluency. For splitting, cells were dissociated using 0.05% trypsin after implementing a washing step using 1 x phosphate buffered saline (PBS). TsA-201 cells were replaced and freshly thawed when they exceeded passage no. 21. For electrophysiology, cells were plated on 10 cm culture dishes and subjected to transient transfections on the following day. Cells were transiently transfected using $Ca^{2+}$-phosphate as previously described (*Ortner et al., 2014*) with 3 µg of α1 subunits, 2 µg of β subunits, 2.5 µg of α2δ1 subunits and 1.5 µg of eGFP to visualize transfected cells.

On the next day, cells were plated onto 35 mm culture dishes that were coated with poly-L-lysine, kept at 30 °C and 5% $CO_2$ and were then subjected to whole-cell patch-clamp experiments after 24–72 hr.

## Primary cell culture of midbrain DA neurons

As described in *Tomagra et al., 2019*, the methods for the primary culture of mesencephalic dopamine neurons from *Substantia nigra* (SN) were adapted from *Pruszak et al., 2009*. Briefly, the ventral mesencephalon area was dissected from embryonic (E15) C57Bl/6 TH-GFP mice (*Matsushita et al., 2002*; *Sawamoto et al., 2001*) that were kept heterozygous via breeding them with C57Bl/6 J mice. HBSS (Hank's balanced salt solution, without $CaCl_2$ and $MgCl_2$), enriched with 0.18% glucose, 1% BSA, 60% papain (Worthington, Lakewood, NJ, United States), 20% DNase (Sigma-Aldrich) was stored at 4 °C and used as digestion buffer. Neurons were plated at final densities of 600 cells per $mm^2$ on Petri dishes. Cultured neurons were used at 8–9 days in vitro (DIV) for current-clamp and voltage-clamp experiments. Petri dishes were coated with poly-L-Lysine (0.1 mg/ml) as substrate adhesion. Cells were incubated at 37 °C in a 5% $CO_2$ atmosphere, with Neurobasal Medium containing 1% pen-strep, 1% ultra-glutamine, 2% B-27, and 2.5% FBS dialyzed (pH 7.4) (as previously described in *Tomagra et al., 2019*).

## Whole-cell patch-clamp recordings in tsA-201 cells

For whole-cell patch-clamp recordings, patch pipettes with a resistance of 1.5–3.5 MΩ were pulled from glass capillaries (Borosilicate glass, catalog #64–0792, Harvard Apparatus, USA) using a micro-pipette puller (Sutter Instruments) and fire-polished with a MF-830 microforge (Narishige, Japan). Recordings were obtained in the whole-cell configuration using an Axopatch 200B amplifier (Axon Instruments, Foster City, CA), digitized (Digidata 1,322 A digitizer, Axon Instruments) at 50 kHz, low-pass filtered at 5 kHz and subsequently analyzed using Clampfit 10.7 Software (Molecular Devices). Linear leak and capacitive currents were subtracted online using the P/4 protocol (20 ms I-V protocol) or offline using a 50 ms hyperpolarizing voltage step from –89 to –99 mV or –119 to –129 mV. All voltages were corrected for a liquid junction potential (JP) of –9 mV (*Lieb et al., 2014*). Compensation was applied for 70–90% of the series resistance.

The pipette internal solution for recordings of Cav2.3 contained (in mM): 144.5 CsCl, 10 HEPES, 0.5 Cs-EGTA, 1 $MgCl_2$, 4 $Na_2ATP$ adjusted to pH 7.4 with CsOH (299 mOsm/kg). The pipette internal solution for recordings of Cav1.3 contained (in mM): 135 CsCl, 10 HEPES, 10 Cs-EGTA, 1 $MgCl_2$, 4 $Na_2ATP$ adjusted to pH 7.4 with CsOH (275 mOsm/kg). The bath solution for recordings of Cav2.3 contained (in mM): 2 $CaCl_2$, 10 HEPES, 170 Choline-Cl and 1 $MgCl_2$ adjusted to pH 7.4 with CsOH. The bath solution for recordings of Cav1.3 contained (in mM): 15 $CaCl_2$, 10 HEPES, 150 Choline-Cl and 1 $MgCl_2$ adjusted to pH 7.4 with CsOH.

Current-voltage (I-V) relationships were obtained by applying a 20 ms long square pulse protocol to various test potentials (5 mV voltage steps) starting from a holding potential of –119 mV or –89 mV (recovery of inactivation). The resulting I-V curves were fitted to the following equation:

$$I = G_{max}\left(V - V_{rev}\right) / \left(1 + \exp\left[-\frac{V - V_{0.5}}{k}\right]\right)$$

where I is the peak current amplitude, $G_{max}$ is the maximum conductance, V is the test potential, $V_{rev}$ is the extrapolated reversal potential, $V_{0.5}$ is the half-maximal activation voltage, and k is the slope factor. The voltage dependence of $Ca^{2+}$-conductance was fitted using the following Boltzmann relationship:

$$G = G_{max} / \left(1 + \exp\left[-\frac{V - V_{0.5}}{k}\right]\right)$$

The voltage dependence of inactivation was assessed by application of 20 ms test pulses to the voltage of maximal activation ($V_{max}$) before and after holding the cell at various conditioning test potentials for 5 s (30 s inter-sweep interval; 10 mV voltage steps; holding potential –119 mV). Inactivation was calculated as the ratio between the current amplitudes of the 20 ms test pulses. Steady-state inactivation parameters were obtained by fitting the data to a modified Boltzmann equation:

$$I = \left(1 - I_{ni}\right) / \left(1 + \exp\left[\frac{V - V_{0.5,\ inact}}{k_{inact}}\right] + I_{ni}\right)$$

where $V_{0.5, inact}$ is the half-maximal inactivation voltage, $k_{inact}$ is the inactivation slope factor and $I_{ni}$ is the fraction of non-inactivating current in steady-state.

The amount of inactivation during a 5 s depolarizing pulse from a holding potential of –119 mV to the $V_{max}$ was quantified by calculating the remaining current fraction after 50, 100, 250, 500, 1000, and 5000 ms. Recovery from inactivation was determined by 10 ms test pulses to $V_{max}$ at different time-points (in s: 0.001, 0.003, 0.01, 0.03, 0.1, 0.3, 0.6, 1, 1.5, 2, 3, 4, 6, 8, 10, 15, 20) after a 1 s conditioning pulse to $V_{max}$ (holding potential –89 mV). Window current was determined by multiplying mean current densities by fractional currents form steady-state inactivation curves to obtain the fraction of available channels at a given potential as described previously (*Hofer et al., 2020*). The SN DA regular pace-making command voltage protocol obtained from an identified TH$^+$ SN DA neuron in a mouse brain slice (male, P12) and the SN DA burst firing protocol were generated as previously described (*Ortner et al., 2017*). Cells were perfused by an air pressure-driven perfusion system (BPS-8 Valve Control System, ALA Scientific Instruments) with bath solution and a flow rate of 0.6 ml/min. For $Cd^{2+}$-block, cells were perfused with 100 µM $Cd^{2+}$ to achieve full block, followed by wash-out with bath solution. A complete exchange of the solution around the cell was achieved within <50 ms. All experiments were performed at room temperature (~22 °C).

## Voltage- and current-clamp recordings in cultured midbrain DA neurons

Macroscopic whole-cell currents and APs were recorded using an EPC 10 USB HEKA amplifier and Patchmaster software (HEKA Elektronik GmbH) following the procedures described previously (*Baldelli et al., 2005*; *Gavello et al., 2018*). Traces were sampled at 10 kHz and filtered using a low-pass Bessel filter set at 2 kHz. Borosilicate glass pipettes (Kimble Chase life science, Vineland, NJ, USA) with a resistance of 7–8 MΩ were used. Uncompensated capacitive currents were reduced by subtracting the averaged currents in response to P/4 hyperpolarizing pulses. Off-line data analysis was performed with pClamp 10.0 software for current clamp recordings. $Ca^{2+}$ currents were evoked by applying a single depolarization step (50 ms duration), from a holding of –70 mV to 0 mV. Fast capacitive transients due to the depolarizing pulse were minimized online by the patch-clamp analog compensation. Series resistance was compensated by 80% and monitored during the experiment.

For current-clamp experiments the pipette internal solution contained in mM: 135 gluconic acid (potassium salt: K-gluconate), 10 HEPES, 0.5 EGTA, 2 $MgCl_2$, 5 NaCl, 2 ATP-Tris and 0.4 Tris-GTP (*Tomagra et al., 2019*). For voltage-clamp recordings the pipette internal solution contained in mM: 90 CsCl, 20 TEA-Cl, 10 EGTA, 10 glucose, 1 $MgCl_2$, 4 ATP, 0.5 GTP and 15 phosphocreatine adjusted to pH 7.4. The extracellular solution for current/voltage-clamp recordings (Tyrode's solution) contained in mM: 2 $CaCl_2$, 10 HEPES, 130 NaCl, 4 KCl, 2 $MgCl_2$, 10 glucose adjusted to pH 7.4. Patch-clamp experiments were performed using pClamp software (Molecular Devices, Silicon Valley, CA, United States). All experiments were performed at a temperature of 22–24°C. Data analysis was performed using Clampfit software. To study the contribution of Cav2.3 channels to the total $Ca^{2+}$ current, cells were perfused with recording solution (containing in mM: 135 TEA, 2 $CaCl_2$, 2 $MgCl_2$, 10 HEPES, 10 glucose adjusted to pH 7.4) complemented with 300 nM TTX and 3 µM ISR to block voltage-dependent $Na^+$ and L-type $Ca^{2+}$ channels. SNX-482 (100 nM) was used in current- and voltage-clamp experiments. Furthermore, kynurenic acid (1 mM), 6,7-dinitroquinoxaline-2,3-dione (DNQX) (20 µM) and picrotoxin (100 µM) were present in the extracellular solution for current- and voltage-clamp experiments.

## Whole cell voltage-clamp recordings of SN DA neurons in ex vivo brain slices

Whole-cell patch-clamp recordings were performed essentially as previously described (*Benkert et al., 2019*). In brief, murine (PN11-13) coronal midbrain slices were prepared in ice-cold ACSF using a VibrosliceTM (Campden Instruments). Chemicals were obtained from Sigma Aldrich unless stated otherwise. ACSF contained in mM: 125 NaCl, 25 NaHCO$_3$, 2.5 KCl, 1.25 $NaH_2PO_4$, 2 $CaCl_2$, 2 $MgCl_2$ and 25 glucose, and was gassed with Carbogen (95% $O_2$, 5% $CO_2$, pH 7.4, osmolarity was 300–310 mOsm/kg). Slices were allowed to recover for 30 min at room temperature (22°C–25°C) before use for electrophysiology. Recordings were carried out in a modified ACSF solution containing in mM: 125 NaCl, 25 NaHCO$_3$, 2.5 KCl, 1.25 $NaH_2PO_4$, 2.058 $MgCl_2$, 1.8 $CaCl_2$, 2.5 glucose, 5 CsCl, 15 tetraethylammonium, 2.5 4-aminopyridine, 600 nM TTX (Tocris), 20 µM CNQX (Tocris), 4 µM SR 95531 (Tocris) and 10 µM DL-AP5 (Tocris), pH adjusted to 7.4, osmolarity was 300–315 mOsm/kg.

Data were digitalized with 2 kHz, and filtered with Bessel Filter 1: 10 kHz; Bessel Filter 2: 5 kHz. All recordings were performed at a bath temperature of 33 °C±1. Patch pipettes (2.5–3.5 MΩ) were filled with internal solution containing in mM: 180 N-Methyl-D-glucamine, 40 HEPES, 0.1 EGTA, 4 MgCl$_2$, 5 Na-ATP, 1 Lithium-GTP, 0.1% neurobiotin tracer (Vector Laboratories); pH was adjusted to 7.35 with H$_2$SO$_4$, osmolarity was 285–295 mOsm/kg. Neurons were filled with neurobiotin during the recording, fixed with a 4% PFA solution and stained for tyrosine hydroxylase (TH; rabbit anti-TH, 1:1000, Cat#: 657012, Merck Millipore) and neurobiotin (Streptavidin Alexa Fluor conjugate 647, 1:1000, Cat# S21374, Thermo Fisher Scientific). Only TH and neurobiotin-positive cells were used for the statistical analysis.

Steady-state activation was measured by applying 150 ms depolarizing square pulses to various test potentials (10 mV increments) starting from –90 mV with a 10 s interpulse interval. Holding potential between the pulses was –100 mV. Voltage at maximal Ca$^{2+}$ current amplitude (V$_{max}$) was determined during the steady-state activation recordings. Voltage-dependence of the steady-state inactivation was measured by applying a 20 ms control test pulse (from holding potential –100 mV to V$_{max}$) followed by 5 s conditioning steps to various potentials (10 mV increments) and a subsequent 20 ms test pulse to V$_{max}$ with a 10 s interpulse interval. Inactivation was calculated as the ratio between the current amplitudes of the test versus control pulse. Currents were leak subtracted on-line using the P/5 subtraction. The series resistance was compensated by 60–90%. Data were not corrected for liquid junction potential (–5 mV, measured according to *Neher, 1992*). Midbrain slices were preincubated (bath-perfusion) at least 30 min in selective T-type (10 µM Z941; *Tringham et al., 2012*), L-type (1 µM isradipine, ISR; https://www.guidetopharmacology.org/), N- and P/Q-type (1 µM ω-conotoxin-MVIIC; https://www.guidetopharmacology.org/) or R-type (100 nM SNX-482; https://www.guidetopharmacology.org/, *Newcomb et al., 1998*) Ca$^{2+}$ channel blockers; except Z941, which was kindly obtained from T. Snutch (University of British Columbia, Canada), all Cav blocker were from Tocris.

Steady-state activation and inactivation curves were fitted as described above.

## Perforated patch recordings in SN DA neurons in ex vivo brain slices

Animals were anesthetized with isoflurane (B506; AbbVie Deutschland GmbH and Co KG, Ludwigshafen, Germany) and subsequently decapitated. The brain was rapidly removed and a block of tissue containing the mesencephalon was immediately dissected. Coronal slices (250–300 µm) containing the SN were cut with a vibration microtome (HM-650 V; Thermo Scientific, Walldorf, Germany) under cold (4 °C), carbogenated (95% O$_2$ and 5% CO$_2$), glycerol-based modified artificial cerebrospinal fluid (GACSF; *Ye et al., 2006*) containing (in mM): 250 glycerol, 2.5 KCl, 2 MgCl$_2$, 2 CaCl$_2$, 1.2 NaH$_2$PO$_4$, 10 HEPES, 21 NaHCO$_3$, 5 glucose adjusted to pH 7.2 (with NaOH) resulting in an osmolarity of ~310 mOsm. Brain slices were transferred into carbogenated artificial cerebrospinal fluid (ACSF). First, they were kept for 20 min in a 35 °C 'recovery bath' and then stored at room temperature (24 °C) for at least 30 min prior to recording. ACSF contained (in mM): 125 NaCl, 2.5 KCl, 2 MgCl$_2$, 2 CaCl$_2$, 1.2 NaH$_2$PO$_4$, 21 NaHCO$_3$, 10 HEPES, and 5 Glucose adjusted to pH 7.2 (with NaOH) resulting in an osmolarity of ~310 mOsm.

SN dopaminergic neurons were identified according to their sag component / slow I$_h$-current (hyperpolarization-activated cyclic nucleotide-gated cation current), broad action potentials or post hoc by TH/DAT-immunohistochemistry (*Lacey et al., 1989*; *Neuhoff et al., 2002*; *Richards et al., 1997*; *Figure 6—figure supplement 1I, J*). Biocytin-streptavidin labeling was combined with TH-immunohistochemistry (*Hess et al., 2013*).

After preparation, brain slices were transferred to a recording chamber (~1.5 ml volume) and initially superfused with carbogenated ACSF at a flow rate of ~2 ml/min. During the perforation process, the electrophysiological identification of the neuron was performed in current clamp mode. Afterwards, the ACSF was exchanged for the Ca$^{2+}$ current recording solution which contained in mM: 66.5 NaCl, 2 MgCl$_2$, 3 CaCl$_2$, 21 NaHCO$_3$, 10 HEPES, 5 Glucose adjusted to pH 7.2 (with HCl). Sodium currents were blocked by 1 µM tetrodotoxin (TTX). Potassium currents and the I$_h$ current were blocked by: 40 mM TEA-Cl, 0.4 mM 4-AP, 1 µM M phrixotoxin-2 (Alomone, Cat # STP-710; *Subramaniam et al., 2014*) and 20 mM CsCl. Experiments were carried out at ~28 °C. Recordings were performed with an EPC10 amplifier (HEKA, Lambrecht, Germany) controlled by the software PatchMaster (version 2.32; HEKA). In parallel, data were sampled at 10 kHz with a CED 1401 using Spike2 (version 7) (both Cambridge Electronic Design, UK) and low-pass filtered at 2 kHz with a four-pole Bessel filter. The

liquid junction potential between intracellular and extracellular solution was compensated (12 mV; calculated with Patcher's Power Tools plug-in for Igor Pro 6 (Wavemetrics, Portland, OR, USA)).

Perforated patch recordings were performed using protocols modified from *Horn and Marty, 1988* and (*Akaike and Harata, 1994*). Electrodes with tip resistances between 2 and 4 MΩ were fashioned from borosilicate glass (0.86 mm inner diameter; 1.5 mm outer diameter; GB150- 8 P; Science Products) with a vertical pipette puller (PP-830; Narishige, London, UK). Patch recordings were performed with ATP and GTP-free pipette solution containing (in mM): 138 Cs-methanesulfonate, 10 CsCl$_2$, 2 MgCl$_2$, 10 HEPES and adjusted to pH 7.2 (with CsOH). ATP and GTP were omitted from the intracellular solution to prevent uncontrolled permeabilization of the cell membrane (*Lindau and Fernandez, 1986*). The patch pipette was tip filled with internal solution and back filled with 0.02% tetraethylrhodamine-dextran (D3308, Invitrogen, Eugene, OR, USA) and amphotericin-containing internal solution (~400 µg/ml; G4888; Sigma-Aldrich, Taufkirchen, Germany) to achieve perforated patch recordings. Amphotericin was dissolved in dimethyl sulfoxide (final concentration: 0.2%–0.4%; DMSO; D8418, Sigma-Aldrich) (*Rae et al., 1991*), and was added to the modified pipette solution shortly before use. The used DMSO concentration had no obvious effect on the investigated neurons. During the recordings access resistance ($R_a$) was constantly monitored and experiments were started after $R_a$ was <20 MΩ. In the analyzed recordings $R_a$ was comparable, did not change significantly over recording time, and was not significantly different between the distinct experimental groups. A change to the whole-cell configuration was indicated by a sudden change in $R_a$ and diffusion of tetraethylrhodamine-dextran into the neuron. Such experiments were rejected. GABAergic and glutamatergic synaptic input was reduced by addition of 0.4 mM picrotoxin (P1675; Sigma-Aldrich), 50 µM D-AP5 (A5282; Sigma-Aldrich), and 10 µM CNQX (C127; Sigma-Aldrich) to the ACSF. For inhibition experiments, 100 nM SNX-482 (Alomone, Cat # RTS-500 dissolved in ACSF) or 10 µM nifedipine (Alomone, Cat # N-120 diluted into ACSF from a freshly prepared 10 mM stock solution in DMSO) was bath applied (in ACSF).

## Identification of β-subunit transcripts in identified SN DA and VTA DA neurons

### RNAscope in situ hybridization

In situ hybridization experiments were performed on fresh frozen mouse brain tissue using the RNAscope technology (Advanced Cell Diagnostics, ACD), according to the manufacturer's protocol under RNase-free conditions and essentially as described (*Benkert et al., 2019*). Briefly, 12 µm coronal cryosections were prepared (*Duda et al., 2018*), mounted on SuperFrost Plus glass slides, dried for 1 hr at –20 °C and stored at -80°C. Directly before starting the RNAscope procedure, sections were fixed with 4% PFA for 15 min at 4 °C and dehydrated using an increasing ethanol series (50%, 75%, 100%), for 5 min each. After treatment with protease IV (ACD, Cat# 322336) for 30 min at room temperature, sections were hybridized with the respective target probes for 2 hr at 40 °C in a HybEZ II hybridization oven (ACD). Target probe signals were amplified using the RNAscope Fluorescent Multiplex Detection Kit (ACD, Cat# 320851). All amplifier solutions were dropped on respective sections, incubated at 40 °C in the HybEZ hybridization oven, and washed twice with wash buffer (ACD) between each amplification step for 2 min each. Nuclei were counterstained with DAPI ready-to-use solution (ACD, included in Kit) and slides were coverslipped with HardSet mounting medium (VectaShield, Cat# H-1400) and dried overnight. Target probes were either obtained from the library of validated probes provided by Advanced Cell Diagnostics (ACD) or self-designed in cooperation with ACD to specifically discriminate between two splice variants of β2, namely β2a and β2e. Target probes (RNAscope assays) used for analysis were as follows: Tyrosine hydroxylase (TH) (*Th*), Cat No. (ACD) 317621, Assay target region: 483–1,603 of NM_009377.1; β2a (*Cacnb2*), Cat No. (ACD) 590951-C2, Assay target region: 2–296 of XM_011238946.2; β2e (*Cacnb2*), Cat No. (ACD) 823221-C2, Assay target region: 2–285 of NM_001309519.1.

Target genes, visualized with Atto550 fluorophore, were co-stained with Tyrosine hydroxylase (TH), visualized with AlexaFluor488, as a marker for dopaminergic neurons. The gene peptidyl-prolyl isomerase B (PPIB) was used as positive control.

Fluorescent images of midbrain sections were acquired by a Leica CTR6 LED microscope using a Leica DFC365FX camera as z-stacks, covering the full depth of cells at ×63 magnification. Z-stacks were reduced to maximum intensity Z-projections using Fiji (http://imagej.net/Fiji) and images were

analyzed by utilizing a custom-designed algorithm (Wolution, Munich, Germany). The algorithm delineates cell shapes according to the TH marker gene signal and quantifies the number and the area of target-mRNA-derived fluorescence dots. Target probe hybridization results in one fluorescent dot for each mRNA molecule, allowing determination of target-mRNA molecule numbers, independent from fluorescence-intensities.

## Multiplex-nested PCR, qualitative and quantitative PCR analysis in individual laser-microdissected DA neurons

Cryosectioning, laser-microdissection, reverse transcription: Cryosectioning, UV-laser microdissection (UV-LMD) and reverse transcription were carried out similarly as previously described in detail (*Benkert et al., 2019*; *Duda et al., 2018*; *Gründemann et al., 2011*; *Liss, 2002*). Briefly, coronal 12 µm mouse midbrain cryosections were cut with a cryomicrotome CM3050 S (Leica), mounted on PEN-membrane slides (Mirodissect), stained with a cresyl-violet (CV) ethanol solution and fixed with an ascending ethanol series. UV-LMD of SN dopaminergic neurons from cresyl-violet-stained midbrain sections from juvenile mice was carried out using an LMD7000 system (Leica Microsystems). Reverse transcription was carried out directly without a separate RNA isolation step in a one-tube procedure.

All cDNA samples were precipitated as described (*Liss, 2002*). Qualitative multiplex-nested PCR and quantitative qPCRs were carried out essentially as described (*Benkert et al., 2019*; *Duda et al., 2018*; *Gründemann et al., 2011*). Briefly, qualitative multiplex-nested PCR was performed with the GeneAmp PCR System 9700 (Thermo Fisher Scientific) using 1/3 of each individual cDNA sample, corresponding to ~3 SN DA neurons, for marker gene expression analysis: tyrosine hydroxylase (*Th*) as a marker for dopaminergic midbrain neurons, the glutamic acid decarboxylase isoforms $Gad_{65}$ and $Gad_{67}$ as markers for GABAergic neurons, the glial fibrillary acidic protein (*Gfap*) as a marker for astroglia cells and calbindin (*Calb1*) d28k (CBd28k), that is strongly expressed only in less vulnerable dopaminergic midbrain neurons. Only cDNA pools expressing the correct marker gene profile (i.e., *Th*-positive, *Gad*-, *Gfap*-, *CBd28k*-negative) were further analyzed via qualitative and quantitative PCR. Quantitative PCR for β2 (*Cacnb2*) splice variant expression analysis was performed with the QuantStudio 3 System (Thermo Fisher Scientific) using ~3 SN DA neurons. Qualitative PCR products were analyzed in a QIAxcel Advanced System (Qiagen).

Details of multiplex PCR (outer) and nested PCR (inner) primers for qualitative PCR are shown in *Supplementary file 5*.

For Cav2.3 (*Cacna1e*) splice variant analysis via qualitative RT-PCR, two cDNA fragments (mCav2.3 II-III loop and mCav2.3 C-terminus) covering the three respective splice sites were amplified in a duplex PCR, followed by two individual nested PCR reactions. PCR conditions: 15 min 95 °C; 35 cycles: 30 s 94 °C; 1 min 61 °C; 3 min 72 °C; 7 min 72 °C for duplex PCR; and 3 min 94 °C; 35 cycles: 30 s 94 °C; 1 min 61 °C; 1 min 72 °C; 7 min 72 °C for nested PCRs. Primer sequences are given in *Supplementary file 2*. Outer duplex primer pairs were chosen using Oligo7.60 software (possible PCR amplicon sizes for mCav2.3 II-III loop: 816 bp, 795 bp and 759 bp; for mCav2.3 C-terminus: 770 bp and 641 bp). For the nested PCRs, primer sequences from *Weiergräber et al., 2005* were used (possible PCR amplicon sizes for mCav2.3 II-III 980 loop: 420 bp, 399 bp and 363 bp; for mCav2.3 C-terminus: 498 bp and 369 bp). Nested PCR products were separated in a 4% agarose gel (MetaPhor agarose, Biozym), and expressed splice variants in individual SN DA neurons were identified according to the amplicon sizes of the nested PCR products (cDNA from mouse whole-brain tissue was used as positive control) as follows (II-III-loop+C-terminus): Cav2.3a: 363+369 bp; Cav2.3b: 399+369 bp; Cav2.3c: 420+369 bp; Cav2.3d. 420+498; Cav2.3e: 363+498 bp; Cav2.3f: 399+498 bp.

Quantitative real time PCR (qPCR) was performed with the QuantStudio 3 System (Thermo Fisher Scientific). qPCR assays (TaqMan), were marked with a 3′ BHQ (black hole quencher) and 5′ FAM (Carboxyfluorescein). TaqMan assays were carefully established and performance was evaluated by generating standard curves, using defined amounts of cDNA (derived from midbrain tissue mRNA), over four magnitudes of 10-fold dilutions as templates, in at least three independent experiments, as described (*Duda et al., 2018*; *Liss, 2002*).

Details of all qPCR assays (TaqMan), and standard curve parameters used for analysis are provided in *Supplementary file 4*.

Relative qPCR quantification data are given as cDNA amount [pg/cell] with respect to midbrain-tissue cDNA standard curves, calculated according to the following formula, and were normalized

to the respective cell size by dividing respective expression values to the corresponding area of the individual microdissected neurons.

$$DNA\ amount\ per\ cell\ \left[\frac{pg}{cell}\right] = \frac{S^{\left[\left(Ct - Y_{intercept}\right)/slope\right]}}{No_{cells} \cdot cDNA\ fraction}$$

With S=serial dilution factor of the standard curve (i.e. 10), No$_{cells}$ = number of SN DA neurons per sample (i.e. 10 here), cDNA fraction = fraction of the cDNA reaction used as template in the qPCR reaction (i.e. 5/17) and the Y-intercept and slope of the relative standards (see *Supplementary file 4*).

β2 (*Cacnb2*) TaqMan qPCR assay and standard curve information, used for relative qPCR-based transcript quantification: Note that the amplicon is very small (70 bp). The probe is 5'-FAM (6-carboxyfluorescein) and 3'-NFQ (non-fluorescent quencher) labelled. The assay Mm01333550_m1 (assay-ID) was used (assay location 583) spanning exon-boundary 4–5 of the mouse β2 gene (*Cacnb2*). Assay parameters were as follows: threshold for analysis: 0.05, Y-intercept of standard curve: 43.5±0.2, slope: –3.43±0.04, R$^2$: 1.0±0.0: means ± SEM, n=5.

## Statistics

Data were analyzed using Clampfit 10.7 (Axon Instruments), Microsoft Excel, SigmaPlot 14.0 (Systat Software, Inc), and GraphPad Prism 5, 7.04 or 8 software (GraphPad software, Inc). Data were analyzed by appropriate statistical testing as indicated in detail for all experiments in the text, figure and table legends. Statistical significance was set at <0.05. Brain slice patch-clamp data were also analyzed with FitMaster (v2 × 90.5, HEKA Elektronik). RNAscope and single-cell RT-qPCR data data were analyzed by Fiji (https://imagej.net/Fiji), QuantStudio Design and Analysis Software (Applied Biosystems) and GraphPad Prism 7.04. All values are presented as mean ± SEM (95% confidence interval) for the indicated number of experiments (n) from N independent experiments (biological replications) in the text and Figures unless stated otherwise.

## Acknowledgements

We thank Dr. Veit Flockerzi for cDNA of β2 splice variants, and Jennifer Müller and Gospava Stojanovic for expert technical assistance.

## Additional information

### Competing interests

Anita Siller: The authors declare that they have no financial and non-financial competing interests. The other authors declare that no competing interests exist.

### Funding

| Funder | Grant reference number | Author |
| --- | --- | --- |
| Austrian Science Fund | P27809 | Jörg Striessnig |
| Tyrolean Science Fund | UNI-0404/2345 | Nadine Jasmin Ortner |
| Italian Miur | 2015FNWP34 | Emilio Carbone |
| Compagnia di San Paolo | CSTO165284 | Emilio Carbone |
| Austrian Science Fund | P35087 | Nadine Jasmin Ortner |
| Hamburg Institute for Advanced Study | Research Fellowship | Birgit Liss |
| Austrian Science Fund | CavX-DOC 30 doc.fund | Jörg Striessnig |
| Austrian Science Fund | P35722 | Jörg Striessnig |
| German Research Foundation | LI-1745/1, GRK1789 | Birgit Liss |

| Funder | Grant reference number | Author |
|---|---|---|
| Alfried Krupp von Bohlen und Halbach Foundation | | Birgit Liss |

The funders had no role in study design, data collection and interpretation, or the decision to submit the work for publication.

## Author contributions

Anita Siller, Conceptualization, Data curation, Formal analysis, Visualization, Writing – original draft, Writing – review and editing; Nadja T Hofer, Giulia Tomagra, Simon Hess, Julia Benkert, Aisylu Gaifullina, Desiree Spaich, Johanna Duda, Christina Poetschke, Peter Kloppenburg, Data curation, Formal analysis, Writing – review and editing; Nicole Burkert, Data curation, Writing – review and editing, Formal analysis; Kristina Vilusic, Data curation, Writing – review and editing; Eva Maria Fritz, Methodology, Writing – review and editing; Toni Schneider, Resources, Writing – review and editing; Birgit Liss, Supervision, Writing – review and editing, Formal analysis, Funding acquisition; Valentina Carabelli, Formal analysis, Supervision, Writing – review and editing; Emilio Carbone, Formal analysis, Supervision, Funding acquisition, Writing – review and editing; Nadine Jasmin Ortner, Conceptualization, Formal analysis, Supervision, Funding acquisition, Methodology, Writing – original draft, Writing – review and editing; Jörg Striessnig, Conceptualization, Supervision, Funding acquisition, Writing – original draft, Project administration, Writing – review and editing

## Author ORCIDs

Peter Kloppenburg http://orcid.org/0000-0002-4554-404X
Emilio Carbone http://orcid.org/0000-0003-2239-6280
Nadine Jasmin Ortner http://orcid.org/0000-0003-3882-3283
Jörg Striessnig http://orcid.org/0000-0002-9406-7120

## Ethics

All animal experiments and procedures were performed in strict accordance with the European Community's Council Directive 2010/63/UE and approved by the Italian Ministry of Health and the Local Organism responsible for animal welfare at the University of Torino (authorization DGSAF 0011710-P-26/07/2017) and the local authorities at the University of Ulm (Regierungsprä;sidium Tübingen, Ref: 35/9185.81-3; Reg. Nr. o.147) and University of Cologne (LANUV NRW, Recklinghausen, Germany (84-02.05.20.12.254)).

## Decision letter and Author response

Decision letter https://doi.org/10.7554/eLife.67464.sa1
Author response https://doi.org/10.7554/eLife.67464.sa2

## Additional files

### Supplementary files

• Supplementary file 1. Names, cellular functions and assay ID's of endogenous control genes. For details see methods.

• Supplementary file 2. TaqMan assays for β1–4 isoforms and N-terminal β2 splice variants including cDNA specific primer sequences for standard template cloning. Additional SalI and HindIII restriction enzyme sites (see Methods) are underlined; the reverse (rev) primer used for standard template cloning including the HindIII restriction site was the same for β2a-e subunit variants; rev, reverse; fwd, forward.

• Supplementary file 3. Quantitative RT-PCR standard curve parameters. SE-B, SE-Y, standard error of B and Y-intercept; Y-int., Y-intercept (CT value); $R^2$, squared correlation coefficient; LOD, limit of detection (number of transcripts); LOQ, limit of quantification (number of transcripts); E (%), Efficiency in % (E=10–1/slope-1), 100% efficiency corresponds to a slope of –3.32.

• Supplementary file 4. Single cell gene expression data. Data and statistics of genes as indicated for graphs shown in *Figure 2—figure supplement 2A* (middle) and B (right). n represents number of analyzed dopaminergic neurons derived from N individual mice. Significances according to two-way ANOVA followed by Tukey's multiple comparisons test: ***, $P < 0.001$.

• Supplementary file 5. Multiplex PCR (outer) and nested PCR (inner) primers for qualitative PCR. F: forward primer, R: reverse primer

• Supplementary file 6. Voltage-dependence of activation and inactivation of Cav1.3 co-transfected with α2δ1 and different β-subunits in tsA-201 cells. All values are given as mean ± SEM for the indicated number of experiments (n, N=2). Parameters were obtained as described in Materials and Methods from a holding potential of –89 mV using 15 mM $Ca^{2+}$ as the charge carrier. Voltage-dependence of gating: Parameters are as given in legend to *Table 1*. Statistical significance was determined using one-way ANOVA with Bonferroni post-hoc test ($V_{0.5}$, $V_{rev}$, act thresh, $V_{0.5,inact}$, $k_{inact}$, plateau) or Kruskal-Wallis followed by Dunn's multiple comparison test (k). Statistical significances of post hoc tests are indicated for comparison vs. β2a (*, **, ***) or vs. β3 (§, §§, §§§): *** $P<0.001$; ** $P<0.01$; * $P<0.05$. Inactivation time course: The r values represent the fraction of $I_{Ca}$ remaining after 50, 100, 250, 500, 1,000 or 5000 ms during a 5 s pulse to $V_{max}$. Statistical significance was determined using one-way ANOVA with Bonferroni post-hoc test. Statistical significances of post hoc tests are indicated for comparison vs. β2a: *** $P<0.001$; ** $P<0.01$; * $P<0.05$.

• Supplementary file 7. Inhibition of ICa in identified SN DA neurons of adult (12 weeks) mice by 100 nM SNX-482 and 10 μM nifedipine (perforated-patch clamp). For details see results. Experiments were performed as described in detail in the Methods section and *Figure 6—figure supplement 1*.

• Transparent reporting form

• Source data 1. Original images.

## Data availability

All data generated or analyzed during this study are included in the manuscript and supporting files. Raw data have been provided for mean population data shown in figures and tables.

The following previously published datasets were used:

| Author(s) | Year | Dataset title | Dataset URL | Database and Identifier |
|---|---|---|---|---|
| Shin W | 2015 | RNASeq of DA neurons from SNpc and VTA. Dataset posted on 04.03.2015, 16:01 by William Shin | https://doi.org/10.6084/m9.figshare.926519.v1 | figshare, 10.6084/m9.figshare.926519.v1 |
| Aguila et al. | 2021 | RNA Sequencing Identifies Robust Markers of Vulnerable and Resistant Human Midbrain Dopamine Neurons and Their Expression in Parkinson's Disease | https://www.ncbi.nlm.nih.gov/geo/query/acc.cgi?acc=GSE114918 | NCBI Gene Expression Omnibus, GSE114918 |

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
