## [Editor Report]

This study finds that voltage-gated calcium channel auxiliary β2a and β2e splice variants confer gating properties on Cav2.3 channels that enable them to contribute sustained Ca^2+^ influx during pacemaking in substantia nigra dopaminergic neurons. This sustained Ca^2+^ influx may contribute to the selective vulnerability of substantia nigra dopaminergic neurons to neurodegeneration in Parkinson's disease. The work will be of great interest to ion channel biophysicists and neuroscientists interested in mechanisms of neurodegeneration.

---

## [Decision Letter]

**Decision letter after peer review:**

Thank you for submitting your article "Alternative splicing of auxiliary β2-subunits stabilizes Cav2.3 ca^2+^ channel activity in midbrain dopamine neurons" for consideration by *eLife*. Your article has been reviewed by 3 peer reviewers, and the evaluation has been overseen by a Reviewing Editor and Richard Aldrich as the Senior Editor. The reviewers have opted to remain anonymous.

Essential revisions:

1) More direct evidence of the putative role of b2a and b2e in supporting Cav2.3 channels in substantia nigra dopaminergic (SN DA) neurons should be provided. Ideally, this would be provided by knockdown experiments showing reduced pacemaking and CaV2.3 currents in response to inhibiting b2a and b2e expression in SN DA neurons. As an alternative, a proximity ligation assay could be done to at least show assembly of b2 and CaV2.3 in SN DA neurons.

2) As suggested by Reviewer 2 the narrative and organization of the manuscript could be improved for more effective communication. The reviewer makes some good suggestions for reorganizing the manuscript that should be considered.

3) Reviewer 1 raised concerns about the adequacy of pharmacological isolation of CaV2.3 currents in the slice recordings that should be addressed. If possible, the isolated CaV2.3 steady-state activation and inactivation curves should be obtained in the cultured neuron preparation where there seems to be better control of pharmacological block of the different Cav channel isoforms.

*Reviewer #2 (Recommendations for the authors):*

Additional comments and suggestions are:

1) The two paragraphs starting on page 7, lines 165-189 should be moved under the header starting on the current page 7, line 191, as it introduces that section.

2) Order of figures and supplemental figures could be improved to follow a revised narrative and clarity. Note that Supp Figure 1 is referenced in the paper after citing Supp Figure 2 and 3.

3) Figure 1: In Figure 1A, it is unclear that the expanded action potentials are representative of the area highlighted. Please revised. The action potential traces in Figure 1C do not seem to be a good representation of the maximum time-derivative of voltage. Please revise.

4) Figure 2: To appreciate the results better, it will help to include control (no drugs) traces and their comparison with the different components.

5) Consider rearranging the order of Figures 3, 4, 5 and 6. The reviewer's suggestion is to make current Figure 5 the new Figure 3. Figure 4 should remain as such. Current Figure 3 and Figure 6 should be merged into new Figure 5. Alternative, current Figure 3 should become a supplemental figure of new Figure 5 (current Figure 6).

6) Current Supp Figure 3 should be included as part of the main figures, specifically as new panel D in current Figure 4.

7) Current Figure 5: Please replace bar plots with scatter plots to visualize the spread of the data. Panel C of this figure is not mentioned in the main text. Please correct. In fact, this panel could go into a supplemental figure.

8) Figure 7: Labels are very small and difficult to read. Please correct.

9) The Supplemental Methods section should be merged with the regular methods section for a comprehensive overview of the methods in a single section.

10) Data in current Supp Figure 1B do not seem to have sufficient statistical power to support conclusions. Please verify.

*Reviewer #3 (Recommendations for the authors):*

#1 Since the crux of this manuscript is the effect of β subunit variants on contribution of CaV2.3 channel in Parkinson's disease, references about the relative expression of β subunit variants in human brain would be extremely relevant and valuable in putting this paper into the context of human disease.

#2 The author's reference to past exploration of the role of β subunits on CaV2.3 should be further expanded. In particular, the assertion that β2a and β2e subunits likely exhibit incomplete inactivation at voltages positive to threshold is not entirely novel and the authors' description (lines 165-179) is slightly misleading. While the authors appropriately cite Jones et al., and Yasuda et al., as having described β subunit effects on the channels, they seem to imply that these papers did not uncover the incomplete inactivation described in the current manuscript. As currently written, the authors imply that these two papers validate the statement in line 173 that R-type currents fully inactivate at voltages positive to -50mV. In fact, after accounting for the different experimental solutions, both papers demonstrate that these β subunits would likely result in available channels at voltages positive to threshold. Reviewer recommends that this point be further clarified in the text.

#3 Currently, the authors make no distinction between the two forms of inactivation in CaV2.3 channels. The experiments are performed in ca^2+^, such that both voltage- and ca^2+^- dependent inactivation will be present. However, past literature indicates that the β subunit effect is primarily on the voltage inactivation of the channel. This should be clearly described.

#4 In Figure 7, the authors have shown the effect of the β2a subunit on the run-down of CaV2.3 channel. How about the β2e subunit? Does it demonstrate similar properties?

---

## [Author Response]

Essential revisions:1) More direct evidence of the putative role of b2a and b2e in supporting Cav2.3 channels in substantia nigra dopaminergic (SN DA) neurons should be provided. Ideally, this would be provided by knockdown experiments showing reduced pacemaking and CaV2.3 currents in response to inhibiting b2a and b2e expression in SN DA neurons. As an alternative, a proximity ligation assay could be done to at least show assembly of b2 and CaV2.3 in SN DA neurons.

We fully agree with the reviewers that it is a relevant point but we do not see how this would be technically feasible. We ask the reviewers to consider the following:

a. Knockdown experiments in SN DA neurons: We do not see how reduced pacemaking would be a suitable readout for more direct evidence of the role of β2a and β2e in supporting Cav2.3 channels in SN DA neurons. We have already shown that, while SNX-482 reduces pacemaking in cultured embryonic dopamine neurons (this paper), neither Cav2.3 KO nor acute pharmacological Cav2.3 channel inhibition (by SNX-482) in SN DA neurons from adult mouse brain slices reduces pacemaker activity (*Benkert et al., Nature Commun 2019; 1*). However, what is reduced by genetic or pharmacological Cav2.3 inhibition in SN DA neurons is the action potential-triggered ca^2+^ influx (*1*), i.e. the potentially harmful ca^2+^-load in these neurons. Hence, the readout of such an experiment would have to be the activity-dependent somatic ca^2+^ influx. To allow unperturbed activity and intracellular signaling, these experiments need to be carried out in perforated-patch configuration. Even without a knockdown approach, these are already challenging experiments in these highly vulnerable SN DA neurons (but feasible, see *1*). Please consider that we would not expect a reduced bulk Cav- or SNX-482-sensitive Cav current in SN DA neurons due to β2a- and β2e-knockdown, but a rather small shift to more negative potentials of the steady-state Cav current inactivation. To address this, a neuron-selective, in vivo AAV viral based Crispr/CAS9 or siRNA approach would be necessary. At the same time, we do not see how it could be possible to demonstrate the cell-specific knockdown of the β2a-and β2e-subunits in the cells recorded from. In order to identify transfected cells and still be able to carry out electrophysiology, one would need to co-transfect a marker like GFP. But, as we have previously shown, GFP itself affects the activity pattern of SN DA neurons (*Schiemann et al., Nature Neurosci 2011; 2*). Again, it would be hardly possible to detect successful recombination, in successfully transfected neurons, and still be able to record from them. With the help of Larry Zweifel (*Hunker et al., Cell Reports 2020; 3*) we have indeed established such an approach for tyrosine hydroxylase (TH), where, however, successful recombination is visible by loss of TH staining. But unlike for TH, we are not aware of a β2-specific antibody expected to work in immunohistochemistry (see new experiments below). These would be required for β2 knockdown verification using juxtacellular labelling, followed by post-PFA fixation and antibody staining (as we did for other targets e.g. in *Lammel et al., Neuron 2008; 4*). Alternatively, one could carry out e.g. a deep sequencing approach but it is difficult to see how this should be possible in combination with prior brain slice electrophysiology/ca^2+^ imaging.

Independent of the arguments above, we also would like the reviewers to consider that a cell-specific β2a+β2e knockdown would also affect other high-voltage-activated Cav channel currents (e.g. P/Q-, N- and L-type), which must also associate with these β-subunits in SN DA neurons. These effects are likely to interfere with the effects on Cav2.3 currents.

Given all these considerations, we unfortunately feel the suggested more direct electrophysiological proof the reviewer asked for is, at present, not feasible.

b. Proximity ligation assay: Another alternative suggested was "… a proximity ligation assay could be done to at least show assembly of β2 and CaV2.3 in SN DA neurons". This is indeed an option, which we thought we could pursue. Since this requires two antibodies from different species, we were searching for and tested suitable anti-Cav2.3 and anti-β2 antibodies. The only anti-Cav2.3 antibody we identified is the one used by us already in the paper Benkert et al., 2019 (*1)* (Cat# 27225-1-AP, Proteintech). It is raised in rabbits and works very nicely in Western-blots (Author response image 1). We were unable to locate a commercial or non-commercial source for an anti-Cav2.3 antibody from another species. We identified two monoclonal mouse anti-β2 antibodies (antibodies specific for β2a- or β2e-splice variants do not exist) which we tested for specificity.

**Author response image 1. sa2fig1:** Western blot using membrane preparations from tsA201 cells transiently transfected with Cav2.3e α1, β3/β2a and α2δ1 or an empty vector (pUC) and mouse whole brain protein preparations were used as described by us previously (14, 15). Tris‐acetate gel: Primary antibodies: rabbit anti‐Cav2.3 (1:15,000; Proteintech, Cat# 27225‐1‐AP) and mouse anti‐tubulin (1:30,000; Sigma‐Aldrich, Cat# CP06). Secondary antibodies: goat anti‐rabbit IgG (1:40,000; Sigma, Cat# A6545) and goat anti‐mouse IgG (1:8,000; Invitrogen, Cat# 31430). M: Marker PageRuler Plus Prestained Protein Ladder (ThermoScientific, Cat# 26619); M2: Marker Spectra Multicolor High Range Protein Ladder (Thermo Scientific, Cat# 26625).

As shown in Author response image 2, the anti-β2 antibody from Antibodies Inc (Cat# 75-065) weakly recognized β2 (but not β3). However, it also strongly cross-reacted with a ~210 kDa band in HEK-cells as well as a ~140 kDa band (Author response image 2, Tris-acetate gel) in mouse brain membranes (calculated molecular mass of the recombinant β2a subunit construct is 68.2 kDa). We also tested different experimental conditions (e.g. Tris-acetate vs. Tris-glycine gels, blotting with or without SDS in transfer buffer) which, however, still revealed this strong non-specific staining (Author response image 2, Tris-glycine gel).

**Author response image 2. sa2fig2:** Western blot (A: Tris‐acetate gel; B: Tris‐glycine gel) using membrane preparations from tsA201 cells transiently transfected with Cav2.3e α1, β3/β2a and α2δ1 or an empty vector (pUC) and mouse whole brain protein preparationsas described by us previously (1, 2). Primary antibody: mouse anti‐β2 (A: 1:1,000; B: 1:800; Antibodies Inc, Cat# 75‐065), secondary antibody: goat anti‐mouse IgG (A: 1:5,000; B: 1:3,000; Invitrogen, Cat# 31430). M: Marker PageRuler Plus Prestained Protein Ladder (Thermo Scientific, Cat# 26619); M2: Marker Spectra Multicolor High Range Protein Ladder (Thermo Scientific, Cat# 26625).

Author response image 3 shows the results with the anti-β2 antibody from Santa Cruz (Cat# sc-81890). This antibody stained β2 but cross-reacted with β3 as shown with recombinant β-subunits. This was a problem in mouse brain where β3 is much more abundant and therefore it was the only β-subunit stained by this β2-antibody (Author response image 3).

We are not aware of any other suitable anti-β2 or anti-Cav2.3 antibodies. Anti-β2 antibodies from other colleagues were raised in rabbits.

**Author response image 3. sa2fig3:** Western blot using membrane preparations from tsA201 cells transiently transfected with Cav2.3e α1, β3/β2a and α2δ1 or an empty vector (pUC) and mouse whole brain protein preparations as described by us previously (1, 2). Tris‐acetate gel: Primary antibody: mouse anti‐β2 (1:800; Santa Cruz, Cat# sc‐81890), secondary antibody: goat anti‐mouse IgG (1:8,000; Invitrogen, Cat# 31430). M: Marker PageRuler Plus Prestained Protein Ladder (Thermo Scientific, Cat# 26619).

2) As suggested by Reviewer 2 the narrative and organization of the manuscript could be improved for more effective communication. The reviewer makes some good suggestions for reorganizing the manuscript that should be considered.

As mentioned above we considered this important criticism, when re-writing major parts of the manuscript, which was necessary because of additional data from novel experiments, some of them suggested by essential revisions 3 (see below). Our data now show different inactivation properties of SNX-482-sensitive Cav2.3 channels in SN DA cells and cultured midbrain neurons within a voltage-range that can be explained by the different contribution of membrane-bound and cytosolic β-subunits. Therefore, the data in cultured neurons are now presented together with those of SN DA neurons (*Figures 6 and 7*) recorded in brain slices. This is outlined in more detail below.

3) Reviewer 1 raised concerns about the adequacy of pharmacological isolation of CaV2.3 currents in the slice recordings that should be addressed. If possible, the isolated CaV2.3 steady-state activation and inactivation curves should be obtained in the cultured neuron preparation where there seems to be better control of pharmacological block of the different Cav channel isoforms.

The reviewer raised concerns about the isolation of R-type current components. His criticism hits an important point: R-type current is defined as current remaining after pharmacological block of all other Cavs. However, several studies found that R-type current components are still present in some cells even after complete knockout of Cav2.3 α1-subunits (e.g. 5, 6) indicating the contribution of other Cavs to Rtype current. In contrast, SNX-482 at 100 nM concentrations is considered a selective inhibitor of Cav2.3 and therefore isolates only Cav2.3-mediated currents, which we are particularly interested in based on their potential pathophysiological role (1) and for which we provide detailed data for β2-splice variant modulation.

Therefore, we not only studied steady-state activation and inactivation of SNX-482-sensitive currents in cultured neurons (as suggested by the reviewer) but also performed such experiments in SN DA neurons in vital brain slices. These findings now reveal:

i. SNX-482-sensitive currents in SN DA neurons inactivate at positive potentials (*Figure 6, Table 3*) very similar to the R-type currents reported in our original manuscript. These data are in agreement with a substantial fraction Cav2.3 channels being stabilized by β2a and/or β2e subunits and therefore resistant to inactivation during pacemaking. ii. The inactivation voltage-range in cultured DA neurons is more negative than in SN DA neurons (but more positive than expected for β3-association) suggesting that inactivation is indeed fine-tuned in neurons along a range of inactivation voltages. For better comparison, data from cultured DA neurons (*Figure 7*) are now presented together with those of SN DA neurons from brain slices (*Figure 6*) at the end of the result section.

Reviewer #2 (Recommendations for the authors):Additional comments and suggestions are:1) The two paragraphs starting on page 7, lines 165-189 should be moved under the header starting on the current page 7, line 191, as it introduces that section.

In the restructured manuscript, information of the first of these paragraphs is now included in the introduction. The second paragraph is now moved to the beginning of the result section to define our experimental approach. We believe that restructuring the manuscript in this way now makes it much easier to follow.

2) Order of figures and supplemental figures could be improved to follow a revised narrative and clarity. Note that Supp Figure 1 is referenced in the paper after citing Supp Figure 2 and 3.

Suppl. Figure 1 ( = *Figure 1—figure supplement 1*) is now referenced first. The order of figures in the revised manuscript was changed based on additional data previous *Figure 1* is now presented as *Figure 7* together with parts moved into new *Figure 7—figure supplement 1*.

3) Figure 1: In Figure 1A, it is unclear that the expanded action potentials are representative of the area highlighted. Please revised. The action potential traces in Figure 1C do not seem to be a good representation of the maximum time-derivative of voltage. Please revise.

These data have been changed as requested (see revised *Figure 7—figure supplement 1*). In the original version the expanded action potentials were inserted twice, which we believe is redundant. We therefore selected two action potentials from the upper trace *Figure 7—figure supplement 1* (indicated by the asterisks) and show the corresponding expanded trace and phase-plane plots in panel C.

4) Figure 2: To appreciate the results better, it will help to include control (no drugs) traces and their comparison with the different components.

We thank the reviewer for this comment. We did not include such a trace because recordings over this time period were stable and run-down in the absence of drugs was less than 1% under these experimental conditions. This is now mentioned in the figure legend (now *Figure 7*) as follows: "Current rundown in the absence of drugs during 150 s was less than 1% (0.48±0.18%; n=4 cells)."

5) Consider rearranging the order of Figures 3, 4, 5 and 6. The reviewer's suggestion is to make current Figure 5 the new Figure 3. Figure 4 should remain as such. Current Figure 3 and Figure 6 should be merged into new Figure 5. Alternative, current Figure 3 should become a supplemental figure of new Figure 5 (current Figure 6).

We merged old Figures 3 and 6 as suggested into new *Figure 3*. We now start with discussing the biophysical properties of co-expressing different β-subunits (old Figure 4 is now *Figure 1*).

6) Current Supp Figure 3 should be included as part of the main figures, specifically as new panel D in current Figure 4.

These data were moved as suggested (now in *Figure 1* as panel D).

7) Current Figure 5: Please replace bar plots with scatter plots to visualize the spread of the data. Panel C of this figure is not mentioned in the main text. Please correct. In fact, this panel could go into a supplemental figure.

We changed to scatter plot as suggested. Panel C was actually mentioned in the text (4^th^ line below the heading “β-subunit transcripts in mouse SN and VTA" in the old manuscript). It is also mentioned in the text of the revised version.

8) Figure 7: Labels are very small and difficult to read. Please correct.

We increased the labels and, based on this valuable feedback, also increased small labels in Figure 8 (now *Figure 5*).

9) The Supplemental Methods section should be merged with the regular methods section for a comprehensive overview of the methods in a single section.

The sections have been merged as suggested.

10) Data in current Supp Figure 1B do not seem to have sufficient statistical power to support conclusions. Please verify.

These are qualitative control experiments (now shown as *Figure 2—figure supplement 1*) that could have been omitted. Yet, we felt to include them in the manuscript. We now mention (and show in Supplementary file 2) that the number of experiments was actually higher because we additionally performed experiments with a mixture of the specific with mismatching β-subunit DNA at ratios of 1:1, 1:2, and 1:5, and only the 1:10 mix is shown (similar CT values for all combinations).

Reviewer #3 (Recommendations for the authors):#1 Since the crux of this manuscript is the effect of β subunit variants on contribution of CaV2.3 channel in Parkinson's disease, references about the relative expression of β subunit variants in human brain would be extremely relevant and valuable in putting this paper into the context of human disease.

To address this important point, we now include a reference for a recent transcriptomic analysis of laser-captured SN DA neurons form 18 human post mortem brains (Aguila et al.; 13) in the Discussion (page 13, bottom/page 14, top paragraph). The results obtained in this study (45% β4, 30% β2, 15% β3, 8% β1) compare nicely with the data obtained by our RT-qPCR assay in mouse substantia nigra: β4 (~65%) and β2 (~27%) represented the most abundant β-subunit transcripts, followed by β1 and β3 (~5 – 7%).

#2 The author's reference to past exploration of the role of β subunits on CaV2.3 should be further expanded. In particular, the assertion that β2a and β2e subunits likely exhibit incomplete inactivation at voltages positive to threshold is not entirely novel and the authors' description (lines 165-179) is slightly misleading. While the authors appropriately cite Jones et al., and Yasuda et al., as having described β subunit effects on the channels, they seem to imply that these papers did not uncover the incomplete inactivation described in the current manuscript. As currently written, the authors imply that these two papers validate the statement in line 173 that R-type currents fully inactivate at voltages positive to -50mV. In fact, after accounting for the different experimental solutions, both papers demonstrate that these β subunits would likely result in available channels at voltages positive to threshold. Reviewer recommends that this point be further clarified in the text.

We thank the reviewer for pointing out this weakness in citing the literature of our colleagues. We now have completely rewritten this paragraph not only to point out the wide range of SNX-482-sensitive Rtype currents reported in neurons but also clearly state that previous work has already shown that βsubunits tightly control V0.5,inact of Cav2.3 channels and that membrane-anchored β2 subunit splice variants β2a and β2e induce strong positive shifts of V0.5,inact of Cav2.3 channels and slow inactivation

(Introduction: page 4, bottom paragraph; page 5, top paragraph; Results: page 6, bottom paragraph).

#3 Currently, the authors make no distinction between the two forms of inactivation in CaV2.3 channels. The experiments are performed in ca^2+^, such that both voltage- and ca^2+^- dependent inactivation will be present. However, past literature indicates that the β subunit effect is primarily on the voltage inactivation of the channel. This should be clearly described.

We believe that this is a more mechanistic aspect not relevant in the context of a study that tries to reproduce physiological conditions in which both inactivation processes must happen together. However, we now also include this information as follows (page 5, top paragraph): ….the membrane anchored β2 subunit splice variants β2a and β2e induce strong positive shifts of V0.5,inact of Cav2.3 channels and slow the time course of voltage-dependent inactivation (Jones et al., 1998; Pereverzev et al., 2002; Soong et al., 1993; Williams et al., 1994; Yasuda et al., 2004).

#4 In Figure 7, the authors have shown the effect of the β2a subunit on the run-down of CaV2.3 channel. How about the β2e subunit? Does it demonstrate similar properties?

We did not measure run-down of Cav2.3 currents in association with β2e and can therefore not state if the effect would be the same. However, based on the current modulation by both “slow” β-subunits, we would assume a similar effect.

References:

Benkert, J., et al., Cav2.3 channels contribute to dopaminergic neuron loss in a model of Parkinson's disease. Nat Commun, 2019. 10(1): p. 5094.Schiemann, J., et al., K-ATP channels in dopamine substantia nigra neurons control bursting and novelty-induced exploration. Nat Neurosci, 2012. 15(9): p. 1272–80.Hunker, A.C., et al., Conditional Single Vector CRISPR/SaCas9 Viruses for Efficient Mutagenesis in the Adult Mouse Nervous System. Cell Rep, 2020. 30(12): p. 4303–4316.e6.Lammel, S., et al., Unique properties of mesoprefrontal neurons within a dual mesocorticolimbic dopamine system. Neuron, 2008. 57(5): p. 760–73.Sochivko, D., et al., The Ca(V)2.3 Ca(2+) channel subunit contributes to R-type Ca(2+) currents in murine hippocampal and neocortical neurones. J Physiol, 2002. 542(Pt 3): p. 699–710.Wilson, S.M., et al., The status of voltage-dependent calcium channels in α 1E knock-out mice. J Neurosci, 2000. 20(23): p. 8566–71.Chan, C.S., et al., 'Rejuvenation' protects neurons in mouse models of Parkinson's disease. Nature, 2007. 447(7148): p. 1081–6.Dragicevic, E., J. Schiemann, and B. Liss, Dopamine midbrain neurons in health and Parkinson’s disease: Emerging roles of voltage-gated calcium channels and ATP-sensitive potassium channels. Neuroscience, 2015. 284: p. 798–814.Ortner, N.J., et al., Lower Affinity of Isradipine for L-Type ca^2+^ Channels during Substantia Nigra Dopamine Neuron-Like Activity: Implications for Neuroprotection in Parkinson's Disease. J Neurosci, 2017. 37(28): p. 6761–6777.Puopolo, M., E. Raviola, and B.P. Bean, Roles of subthreshold calcium current and sodium current in spontaneous firing of mouse midbrain dopamine neurons. J Neurosci, 2007. 27(3): p. 645–56.Mercuri, N.B., et al., Effects of dihydropyridine calcium antagonists on rat midbrain dopaminergic neurones. Br J Pharmacol, 1994. 113(3): p. 831–8.Guzman, J.N., et al., Robust pacemaking in substantia nigra dopaminergic neurons. J Neurosci, 2009. 29(35): p. 11011–9.Aguila, J., et al., Spatial RNA Sequencing Identifies Robust Markers of Vulnerable and Resistant Human Midbrain Dopamine Neurons and Their Expression in Parkinson’s Disease. Frontiers in Molecular Neuroscience, 2021. 14.Scharinger, A., et al., Cell-type-specific tuning of Cav1.3 Ca(2+)-channels by a C-terminal automodulatory domain. Frontiers in Cellular Neuroscience, 2015. 9: p. 309.Pinggera, A., et al., CACNA1D de novo mutations in autism spectrum disorders activate Cav1.3 Ltype calcium channels. Biol Psychiatry, 2015. 77(9): p. 816–22.